# The social readjustment rating scale: Updated and modernised

Denise Wallace[1]*, Nicholas R. Cooper[1], Alejandra Sel[1], Riccardo Russo[1,2]

**1** Department of Psychology and Centre for Brain Science, University of Essex, Colchester, Essex, United Kingdom, **2** Department of Behavioral and Brain Sciences, University of Pavia, Pavia, Italy

☯ These authors contributed equally to this work.

* dwallace@essex.ac.uk

**Data Availability Statement:** All relevant data are within the paper and its Supporting Information files.

**Funding:** The author(s) received no specific funding for this work.

## Abstract

The Social Readjustment Rating Scale, originally devised in 1967 by Holmes and Rahe, measures the impact of life events stress. At the time, the SRRS advanced its field of research by standardising the impact of stress with a set of independently derived weights called 'life change units' (LCUs) for 43 life events found to predict illness onset. The scale has been criticised for being outdated, e.g. "Mortgage over $10,000" and biased, e.g. "Wife begin or stop work". The aim of this cross-sectional survey study is to update and improve the SRRS whilst allowing backwards compatibility. We successfully updated the SRRS norms/LCUs using the ratings of 540 predominantly UK adults aged 18 to 84. Moreover, we also updated wording of 12 SRRS items and evaluated the impact of demographics, personal experience and loneliness. Using non-parametric frequentist and Bayesian statistics we found that the updated weights were higher but broadly consistent with those of the original study. Furthermore, changes to item wording did not affect raters' evaluations relative to the original thereby ensuring cross-comparability with the original SRRS. The raters were not unduly influenced by their personal experiences of events nor loneliness. The target sample was UK rather than US-based and was proportionately representative regarding age, sex and ethnicity. Moreover, the age range was broader than the original SRRS. In addition, we modernised item wording, added one optional extra item to the end of the scale to evaluate the readjustment to living alone and identified 3 potential new items proposed by raters. Backwards-compatibility is maintained.

## Introduction

The Social Readjustment Rating Scale (SRRS) is a 43-item list of typically experienced life change events commonly used by researchers interested in the impact of stress on health and well-being. It was designed to predict the allostatic load (physiological cost) of the transient social adjustment required when certain life events occur (e.g. marriage, traffic ticket or a loan). It is well-validated and is cited in over 6000, widely varied, scientific publications. For example, it has been used to measure the association between experienced stress and accelerated cognitive ageing [1–4], to measure suicide risk [5] and to evaluate the impact of stress severity on dermatitis [6]. The life events were chosen based on sound empirical evidence [see

**Competing interests:** The authors have declared that no competing interests exist.

references 1–12 cited in [7] that is arguably still relevant today [e.g. 8–11]. Numerous updates to the rating norms and modifications to the scale items have been undertaken [10,12–15] to address validity and reliability concerns [e.g. 16–18], yet many researchers still use the original version [e.g. 1,19,20]. A brief history of the SRRS' development is provided, highlighting reasons for updates and modifications and why these may have failed to persuade researchers to deviate from the original. The proposed updates are then outlined.

The SRRS evolved from the Schedule of Recent Life Events (SRE) [21,22] which captures a broad spectrum of 42 positively and negatively valenced items which require some level of social readjustment, desirable or undesirable, life-changing or minor [23]. Social readjustment refers to the amount and duration of change in one's usual routine resulting from various life events. Holmes and Rahe's SRRS comprises the 42 SRE items plus "Christmas". Holmes and Rahe's [7] SRRS study revealed marked similarity between sub-groups in terms of the relative significance of the life events (e.g. 'marriage' vs. 'death of a spouse'), indicating some level of universal agreement for certain experiences. The primary aim of the SRRS was to improve the precision with which the impact of life events on illness onset was measured. Each item has an averaged weighting based on the estimated magnitude of change assigned by a convenience sample (n = 394) of males (n = 179) and females (n = 215) who varied in age, class, education, marital status, religion and race. The raters were asked to rate the magnitude of social readjustment required for each life event irrespective of the desirability of the event, using all their experience as well as what they had learned to be the case for others, *relative* to the social readjustment needed after marriage. Marriage served as the anchor item with an arbitrary value of 500. The weight for each item was then derived by taking the raters' average weight and dividing by 10. These weights represent 'Life Change Units' (LCU). This set of 43 LCUs provides a set of norms that accompany the SRRS. Social Readjustment Rating Scale respondents would indicate which of the 43 items they have experienced over a certain time-frame (e.g. the previous 12 months). All the LCUs corresponding to the respective items are then summed to produce a total LCU value, which may be used to predict physiological and/or psychological impact for each respondent. For example, a respondent might tick "Death of Spouse" which is 100 LCUs, "Troubles with the boss" (30 LCUs) and "Change in residence" (32 LCUs) giving a total of 162 LCUs. Based on empirical work, Rahe [24] found that a score of about 150 suggested that the respondent would remain healthy over the next 12 months while those falling ill over the same period were typically found to score > 300.

The SRRS and SRE were applauded for adding an objective element to the study of life stress and its impact on health. However it was also argued that the scale was inherently flawed because the event items were not equally well-comprehended by less educated samples [e.g. 25] and the accuracy of event-reporting varied [17]. Furthermore, operationalising "illness" and "life events" are difficult [26]. One review indicated life events likely explained no more than 9% of illness variance [27] and Rahe and colleagues themselves stated that precipitating stressful life events were a necessary but not sufficient antecedent to illness onset [28]. Researchers have sought to address some of these and other concerns as described below.

Muhlenkamp and colleagues [12] noted that the SRRS normative sample did not include those over age 70. They published an extension, providing independent ratings from a sample (n = 41) of 65 to 84 year-olds and modified the instructions by assigning a value of 50 for marriage, rather than 500, to provide a more meaningful and familiar anchor for participants. The study found that raters gave higher ratings for most items relative to the original but there was significant agreement regarding the rank ordering of items. However, these weights were never used in conjunction with any subsequent application of the SRRS by researchers including Miller and Rahe's [13] update, which replicated the characteristics of the original sample. Moreover, no further validation was undertaken of Miller and Rahe's [13] update, which may

have hindered its adoption in future studies. To our knowledge, researchers have sought only to apply the original weights though modifications to the scale items were undertaken on an ad hoc basis. For example Komaroff and colleagues [25] substituted "marital reconciliation with spouse" with "getting back together" as it was more meaningful to the target population.

Hobson and colleagues [14,15] addressed sample and content criticisms of the original SRRS with an extended, modified "Social Readjustment Rating Scale Revised" (SRRS-R). To address the SRRS' outdated and insufficiently representative sample the SRRS-R was based on norms derived from a larger sample (n = 3122), representative of a cross-section of Americans regarding age, race, gender, ethnicity, income and geographical location. To address criticisms around content, Hobson and colleagues asked a 30-member expert panel to add, amend or remove existing items, which produced a 51-item scale. Other criticisms of the original version were that some items can be interpreted as symptoms/outcomes rather than precipitating events—the 'contamination hypothesis', e.g. "Change in sleeping habits", could indicate that a new job, like shift work (precipitating event), has occurred or it could indicate the symptom/outcome of a stressful experience. Some items lack representativeness in modern, multi-cultural societies (e.g. "Christmas") and some items' wording is ambiguous, biased or out-dated (e.g. "Mortgage or loan greater than $10,000"). Using their extended, modified scale, Hobson et al. [14] found that there were significant differences in the way individuals evaluated the stressfulness of different events. On this basis they concluded that using simple unitary weights (occurred vs. not occurred) risked masking these differences and that further work needs to assess the impact of using group-based weights vs. individually derived weights vs. unit weights. Whilst they found that results were statistically significant, effect sizes were very small and ratings were remarkably similar across age, gender and income categories. Their approach validly addressed concerns however the SRRS-R departed notably from the original SRRS negating any opportunity for cross-comparability and, consequently, the SRRS-R has not been incorporated into any subsequent publications to the best of our knowledge.

Around the same time, Scully and colleagues [10] published updated SRRS ratings and addressed 3 content-related criticisms of the SRRS. They assessed the validity of the contamination hypothesis, mentioned previously. In addition, some evidence suggest that undesirable life events would have a stronger stress response than desirable ones [29,30] though not all findings agree [31]. Similarly, uncontrollable life events would have a more potent stress impact than controllable ones (ibid). In phase 1 of Scully and colleagues' study [10], the original SRRS instructions were administered to a random sample of Florida residents (n = 200) whose ratings were used to derive updated weights (LCUs) for all items. In phase 2, another sample completed the SRRS, reporting experienced events a) within the last 12 months and b) ever. They also completed a modified version of the Symptom Checklist-90 which measures stress-related symptoms. A group of university staff and student raters (n = 7) categorised all the SRRS items as desirable, undesirable or neutral. A separate group of student raters (n = 7) categorised the items as either controllable or uncontrollable. Comparisons of symptom reporting were conducted based on these categorisations. Regression analyses revealed that the SRRS in its original form was predictive of stress symptoms. In addition, consistently more variance was explained when regression models included all items than when only respective undesirable/uncontrollable items were included. Thus, including only negative items was found to limit the utility of the SRRS. They also found that symptoms associated with events reported over the last 12 months had greater predictive power, suggesting that the stress impact of life events diminished with passing time. They [10] concluded that "the SRRS is a robust instrument for identifying the potential for stress-related outcomes" (p.875).

Twenty years on from the last attempts to modernise the SRRS, the primary aim of the current study was to update and improve the SRRS without fundamentally changing the scale to

allow for cross-comparison of studies, which may have played a role in previous updates not being incorporated into subsequent versions. Six areas of focus were identified: First, the original weightings are 5 decades old and required updating. Second, biases in item wording were removed. Third, the complete and accurate wording from the raters' version was re-instated. As Holmes and David [32] pointed out: "We regret the decision to save space on the Social Readjustment Rating Scale, because the complete wording is the accurate and more helpful form." (p.30). The SRRS 'rating' questionnaire comprised detailed statements, along with examples in some cases. The actual scale's wording is much simplified. For example:

- Raters assessed: "Major change in usual type and/or amount of recreation". The final SRRS used to evaluate illness onset was simplified to: "Change in recreation".

- Raters assessed: "Minor violations of the law e.g. traffic tickets, jay walking, disturbing the peace". The final version was simplified to: "Minor violations of the law".

Thus, the final version leaves the reader to make assumptions about what 'counts' and what does not, resulting in increased inter-individual differences in responding. This portion of inter-individual variability was reduced by reinstating the 'rater' version of items.

Fourth, information was collected regarding the potential for systematic bias that may have affected the magnitude of weight that raters assigned to each item. Raters were asked to indicate the extent to which their rating was based on their own personal experiences of events. They were also asked how lonely they were and how frequently they felt lonely. Loneliness was chosen for two reasons, firstly as part of the evaluation of a new item, "Single person, living alone", that was added to the end of the scale and secondly, as a proxy for depression which is associated with loneliness and stress [33–36]. Thus, loneliness allowed the evaluation of whether ratings varied based on emotional state at the time of rating.

Fifth, the rater sample was made more representative, proportionately reflecting the demographics within the UK regarding age, gender and ethnicity.

Sixth, the need for new items was considered. A rating was added to the norms set for being single and living alone, with its inclusion as optional at the end of the SRRS. In addition, an opportunity was provided for raters to add an item and its weight, which they believed could improve future work regarding what people find difficult to adjust to at the current time.

In this cross-sectional survey study, an assessment was made of the extent to which the sample's ratings deviated from those of the original, replicating earlier similar analyses. Previous work indicates that this is likely. For example, Miller and Rahe [13] in their update found ratings differed when comparing males and females and married with unmarried individuals. Women's ratings were, on average, 17% higher than those of men. Muhlenkamp et al. [12] measured differences between their elderly sample and the original raters with items categorised into 'family', 'personal', 'work' and 'finance'. A replication of this analysis was undertaken. The extent to which the rank order of items from the updated SRRS agreed with that of the original was also evaluated.

## Materials and method

### Study design

A survey method was used comprising a series of questionnaires administered via Qualtrics in a single, online-only session. The present study broadly replicates that of Holmes and Rahe [7] who recruited a convenience sample of adults aged $\geq$ 18 years to rate a list of 42 life events, using a proportional scaling method.

## Participants

Six hundred and thirty adults aged 18 to 85 accepted the invitation to participate. The sample selection criteria were based on the UK's current age distribution and gender breakdown and England and Wales' ethnicity breakdown published by the ONS [37]. Based on ONS estimates for England and Wales, 84.8% of the population is white. Roughly that proportion of Caucasians was recruited with the remaining proportion comprising non-Caucasian ethnic groups. Regarding sex, a 50/50 split was targeted, reflecting a similar split within the UK population. Ethnic and sex breakdowns were nested within age bands proportioned as per ONS statistics.

Participants were recruited via social media, word-of-mouth, SONA (local university student recruiting platform) and Prolific, an online participant recruitment platform. Participants had to be ≥ 18 years to be included in this study. Within the Prolific platform participants currently located in the UK were selected and anyone who had taken part in any of our previous studies were excluded. No other exclusion criteria were applied. Participants were recruited and data collected from February 2021 to May 2021. Respondents were anonymous; no personally identifiable information was collected. Thus, participants could not be identified during or after data collection. Participants volunteered either without payment, received a small payment or course credits. The study was approved by the University of Essex Faculty of Science and Engineering Ethics Committee (ETH2021-0829). All participants gave written informed consent using an online form, which had to be read and agreed to before they could gain access to the study.

## Measures

**Social Readjustment Rating Questionnaire (SRRQ).**   The updated SRRQ with instructions administered to the rater sample is provided in Appendix 1 (S1 Appendix). For comparison, the original SRRQ instructions are provided in Appendix 2 (S2 Appendix). To reduce potential variations in interpretation, the SRRQ was administered with modified instructions based on those of Muhlenkamp et al. [12] who changed the weight for marriage from 500 to 50 and simplified the instructions themselves. Using marriage (50) as the anchor point, participants were instructed to rate each item from 0 to 100. Some wording was simplified but kept as close to the original as possible, asking participants to draw on their experience and those of others when giving their ratings, as in the original version. Note that in the original SRRQ participants rated 42 items (relative to marriage). The updated SRRQ includes a 43$^{rd}$ item to be rated: 'Single person, living alone'. The outcome variables for the SRRQ were the mean weights assigned to each of 43 items (range: 0–100). The mean ratings were derived by averaging the ratings given across participants for each respective item.

**Social Readjustment Rating Scale 2022 updated.**   The updated SRRS, used in conjunction with the SRRQ ratings, is provided in Appendix 3 (S3 Appendix). In accordance with the SRRQ, the updated SRRS also contains the new item at the end of the scale, 'Single person, living alone'. Consequently, the total number of life change units for a given participant is based on 44 items rather than the original 43. Participants were asked to respond 'yes' or 'no' with the instruction: "Please indicate which of the following events have occurred <u>in your whole life</u>". The order of items were randomised and then presented in the same order across participants. Some subtle updates or clarifications to wording were applied e.g. 'spouse' became 'spouse/life partner'. Due to inflation the monetary value used for loans was removed, as recommended by Holmes and David [32]. The outcome variable was the binary value for each of the items multiplied by the corresponding item 'weight' (life change units). The products were then summed to provide a total life change units score which represents one's life change

intensity [38]. A higher value indicates greater intensity (i.e. a greater level of adaptation to change was needed).

**Invitation to add own item.** In this single-item questionnaire respondents were asked: "If you could add one more item to the list, what would it be?". Participants used the free text box to provide a response or they could leave it blank and continue. In the follow-up question, participants were asked to provide a rating for this item relative to marriage. Valid responses therefore required 2 components: a life event (given in their own words) and a corresponding rating.

**Experiential basis for SRRS ratings.** An instruction was given to measure the extent to which the participants' ratings were based on their own experience: "At the start of this survey you were asked to rate a range of life events by comparing them to marriage. To what extent was your chosen rating based on your own personal experience? Please slide the scale to indicate as best you can how much your rating was based on your own experience from 'not at all based on my own experience' (0) to 'completely based on my own experience' (100)." Appendix 4 (S4 Appendix) provides a copy of this questionnaire. The outcome variable was the value given for each item (range: 0 to 100).

**Loneliness questionnaire.** The ONS recommends the following 4 questions to measure loneliness: "How often do you feel that you lack companionship?", "How often do you feel left out?" and "How often do you feel isolated from others?" with response options of 'hardly ever or never = 1', 'some of the time = 2' and 'often = 3'. Scores for these 3 items are summed (range: 3 to 9). A higher score indicates a greater degree of loneliness. The 4th question asked: "How often do you feel lonely?" with 6 response options ranging from 'often/always' = 1 to 'never' = 5 and 'prefer not to say' = 6 [37]. A lower score indicates a greater level of loneliness. The first 3 questions were taken from the University of California, Los Angeles loneliness scale (UCLA v3) [39] which was adapted to a 3-item scale: R-UCLA [40] as used in English Longitudinal Study of Ageing [33]. The UCLA scale has good reliability (coefficient α range: 0.89 to 0.94) and test re-test reliability (r = 0.73). The scale's reliability and validity was tested on students, teachers, nurses and elderly participants (> 65 years). The R-UCLA has an alpha coefficient of 0.72 with good internal consistency [40]. The final question forms part of the Community Life Survey [41]. Appendix 5 (S5 Appendix) provides a copy of these survey items. Thus, loneliness was measured with two outcome measures: loneliness level as a summed value (range: 3 to 9); loneliness frequency as a single-item value (range: 1 to 5).

## Procedure

Participants read the information sheet, accepted the invitation to take part, gave online informed consent by completing a check-list then provided biographical details, namely age, ethnicity, gender, religion, relationship status and employment status (see Results). They were presented with the following in sequential order: the updated SRRQ (rating questionnaire), updated SRRS, new item with corresponding rating, personal experience questionnaire, 4-item loneliness questionnaire.

## Statistical analysis

Descriptive statistics are provided for all items. Data were analysed using parametric and/or non-parametric analyses alongside equivalent Bayesian comparisons, depending on whether distributions were normal or skewed. Where any disparity existed between frequentist and Bayesian results, conclusions were based on the Bayes factors (BF), which is not subject to the stopping rule as with the frequentist method. Bayesian analyses were conducted in JASP 0.16.2.0 [42]. Sensitivity analyses for BFs for between-groups comparisons were also conducted to ensure that expected effect sizes were robust.

Numerous previous SRRS studies used geometric means. However, the original SRRS weights were derived using arithmetic weights therefore these were used (see the Results section). For frequentist analyses, alpha levels at < .05 were applied to control for type 1 error. To be comparable to Miller and Rahe (1997), 99% confidence intervals were used.

Gpower 3.1.9.7 [43] indicated that for correlational analyses a sample size of 319 would be required to achieve a power of .80 assuming a small effect size (0.2); for chi square analyses with 1 degree of freedom it was 197. However, an independent samples t-test comparison would require a sample size of 788 and a Mann-Whitney U test would require 824 participants. Due to budget and time constraints, we set the N value at the top end of these constraints. For the Bayesian analyses sensitivity analyses were also performed to evaluate the robustness of BFs. In JASP, the default prior effect size is modelled on a Cauchy distribution scaled to $\frac{\sqrt{2}}{2}$ which captures small effects efficiently but has fatter tales than the normal distribution to also capture larger effects. JASP performs sensitivity analyses automatically for parametric tests (i.e. independent t-tests) by varying the scale. Thus, $\frac{\sqrt{2}}{2}$ is substituted with a wide (*Chaucy prior scale = 1.00*) and then an ultra-wide (*Chaucy prior scale = $\sqrt{2}$*) scale. Bayes factors are robust when they remain stable under these varied conditions. When non-parametric between-groups tests are performed, a similar approach will be taken by manually performing these additional BF calculations using the different scaling parameters. The latter mitigates the limitation posed by the smaller-than-needed sample size for between-groups comparisons.

The open-ended item ('Invitation to add own item' outlined earlier) was analysed by a simple frequency method where the number of times a particular event was given was counted as '1'. Only events with an accompanying rating were counted.

## Results

Six hundred and thirty respondents were recruited and consented to take part in the study. Ninety of 630 respondents logged off from the study part-way and their data could not be used. Of the remaining 540 respondents, all completed the study in full apart from 5 who completed most of the study ($\geq 90\%$). As part of the informed consent, all participants had agreed that any data collected up to the point of withdrawal may be used. These 5 respondents' data were therefore retained. None provided data for the loneliness questionnaire, which was the last item of the study. Four of the 5 respondents rated the SRRQ and gave responses to the SRRS but logged off without completing the remaining items (personal experience, selecting own item, loneliness). Two further participants did not answer the question regarding loneliness frequency. Thus for loneliness frequency, 7 respondents' data are missing. Two participants identified as 'gender fluid'. For clarity, participants were therefore grouped by biological sex (male vs. female) and the gender-fluid participants' data were excluded from all gender/sex-based analyses. Analyses were conducted with all available data using list-wise or pair-wise deletion, as appropriate. Sample sizes are given for each table.

Descriptive statistics for demographic details are given in Table 1. There were 453 Prolific participants (84%), SONA (8%) and social media/word-of-mouth (8%). Eighty-seven percent of the sample were British, 4% were EU nationals, 2% were USA nationals and 7% were from other countries. Most (95%) reported English as their first language.

Wording was adjusted/modernised on 12 items of the rating questionnaire, as shown in Table 2. To assess whether this may have caused those weights to change by more than the unchanged items a Mann-Whitney U test was conducted on the difference scores (new minus original weight) by wording-changed vs. wording-unchanged items. The difference was not statistically significant (p > .685) ($Mdn_{changed}$ 10.2 vs. $Mdn_{unchanged}$ 8.8) though the equivalent

**Table 1. Descriptive statistics for biodemographic details of the total sample.**

| | | mean (min-max) | median (IQR) | n* (%) |
|---|---|---|---|---|
| Age | < 30 years | 23.08 (18–29) | 22 (20–27) | 116 (21.5) |
| | 30 to 60 years | 45.18 (30–60) | 45 (37–53) | 291 (53.9) |
| | > 60 years | 69.81 (61–84) | 70 (65–75) | 133 (24.6) |
| Gender | female | | | 308 (57) |
| | male | | | 230 (42.6) |
| | gender-fluid | | | 2 (0.4) |
| Education | years | 15.31 (12–21) | 15 (14–17) | 540 (100) |
| Relationship status | married | | | 236 (43.7) |
| | long-term relationship | | | 107 (19.8) |
| | in a relationship | | | 30 (5.6) |
| | separated | | | 5 (0.9) |
| | divorced | | | 21 (3.9) |
| | widowed | | | 11 (2) |
| | life partner died | | | 5 (0.9) |
| | single | | | 125 (23.2) |
| Ethnicity | white | | | 455 (84.3) |
| | mixed race (all) | | | 23 (4.3) |
| | Asian (southern/southeastern Asia) | | | 24 (4.4) |
| | Chinese (east Asian) | | | 7 (1.3) |
| | black (any region) | | | 31 (5.7) |
| Religion | no religion | | | 266 (49.3) |
| | Christian | | | 234 (43.3) |
| | Buddhist | | | 4 (0.7) |
| | Hindu | | | 5 (0.9) |
| | Jewish | | | 6 (1.1) |
| | Muslim | | | 17 (3.2) |
| | Sikh | | | 5 (0.9) |
| | any other religion | | | 3 (0.6) |
| Employment status | full-time or part-time employed | | | 348 (64.4) |
| | currently unemployed, looking for work | | | 14 (2.6) |
| | long-term sick or disabled | | | 11 (2) |
| | looking after home or family | | | 28 (5.2) |
| | retired | | | 97 (18) |
| | have never worked | | | 2 (0.4) |
| | student (p/t or f/t) and currently unemployed | | | 39 (7.2) |
| | other | | | 1 (0.2) |

*N = 540.

BF of 0.35 was within the anecdotal range, approaching the null, suggesting that participants' ratings were unlikely to vary with the wording changes.

## SRRS ratings then and now: A comparison with Holmes & Rahe (1967)

The original scale had a score-range of 0 to 1466, while the newly weighted version's range extended to 1871, increasing the total by 405 life change units (LCUs). Of the original 42 event items rated, 39 increased and 4 decreased. Of the items that increased, 3 items increased by $\geq 25$ LCUs relative to the original scale: 'Foreclosure/repossession on mortgage or loan'

**Table 2. Changed items.**

|  | Original item wording | New item wording |
|---|---|---|
| 1. | Death of spouse | Death of a spouse or life partner |
| 2. | Minor violations of the law (e.g. traffic ticket, jay walking, disturbing the peace) | Minor violations of the law (e.g. traffic ticket, disturbing the peace) |
| 3. | Pregnancy | Pregnancy (either yourself or being the father) |
| 4. | Gaining a new family member (e.g. through birth, adoption, oldster moving in, etc.) | Gaining a new family member (e.g. through birth, adoption, grandparent moving in, etc.) |
| 5. | Marital separation from mate | Marital separation |
| 6. | Major change in church activities (e.g. a lot more or a lot less than usual) | Major change in religious activities (e.g. a lot more or a lot less than usual) |
| 7. | Marital reconciliation with mate | Marital reconciliation |
| 8. | Being fired from work | Losing your job (redundancy, dismissal, etc.) |
| 9. | Major change in the number of arguments with spouse (e.g. either a lot more or a lot less than usual regarding child-rearing, personal habits, etc.) | Major change in the number of arguments with spouse or life partner (e.g. either a lot more or a lot less than usual regarding child-rearing, personal habits, etc.) |
| 10. | Spouse begins or stops working outside the home | Spouse or life partner begins or stops working |
| 11. | Taking on a mortgage greater than $10,000 (e.g. purchasing a home, business, etc.) | Taking on a mortgage or loan for a major purchase (e.g. purchasing a home, business, etc.) |
| 12. | Taking on a mortgage or loan less than $10,000 (e.g. purchasing a car or furniture, paying for college fees, etc.) | Taking on a loan for a lesser purchase (e.g. purchasing a car or furniture, paying for college fees, etc.) |

($62_{new}$ vs. $30_{original}$), 'Death of a close friend' ($64_{new}$ vs. $37_{original}$) and 'Pregnancy' ($65_{new}$ vs. $40_{original}$). When including the new 43rd rater item, 'Single person, living alone', the range increases to 1909 (1871+38). A Mann-Whitney U test found that total LCUs were, on average, higher in the current (Mdn 40.1, n = 43) relative to the original scale (Mdn 29, n = 43) (z = -2.807, p = .005, r = .3). The Bayesian Mann-Whitney U test supported this finding with substantial evidence (BF = 6.22). Bayesian sensitivity analyses are provided in Appendix 6 (S6 Appendix) and indicate that applying different Bayesian priors did not affect the outcome, therefore the reported BFs are reliable. A Kendall's tau correlation coefficient was conducted to evaluate the level of agreement between the two scales. This revealed that the original weights were strongly, positively associated with the new weights (r = .751, p < .001). The corresponding Bayesian Kendall's tau provided decisive evidence for this finding (BF > 100), suggesting that the respondents in the new scale and the original rating sample were comparable in the hierarchy of change for their evaluations. Table 3 presents the descriptive statistics and rank order of the SRRS items for the original and new scale weights. The table provides the arithmetic means with their standard errors and 99% confidence intervals, plus the geometric means and median values. The events are ordered by the rank of absolute change in number of LCUs, from highest to lowest (1 to 43) to provide a visual comparison of change between original and new weights. Regarding how consistent participants' ratings were for each item, it was observed that the range of ratings spanned the full range of 0 to 100 on most items (38/43). The magnitude of interquartile ranges (IQR) and 99% bias-corrected and accelerated (BCa) confidence intervals for each of the 43 items were therefore inspected to better ascertain consensus of ratings. Table 3 shows that the largest IQR magnitude was 40, which was for 'Outstanding personal achievement' and 'Retirement from work'. Next largest were 'Gaining a new family member' (39), 'Death of a close family member' (35), 'Son or daughter leaving home' (35), 'Losing your job' (35), 'Taking on a mortgage or loan for a major purchase' (35), 'Major business readjustment' (35) and 'Single person, living alone' (35). The smallest IQR was for

**Table 3. Present study vs. original weights and rank order of SRRS events\*.**

| Event | Holmes & Rahe | | Present Study | | | | | Difference Score | |
|---|---|---|---|---|---|---|---|---|---|
| | original rank | weight | present rank | Mean (SE)[a] | 99% CI[b] | Geometric Mean[c] | Median (IQR) | present— original weight | rank: absolute change[d] |
| Foreclosure/repossession on mortgage or loan | 21 | 30 | 9 | 62.39 (1.07) | 59.65, 65.09 | 32.71 | 40 (25–60) | 32.4 | 1 |
| Death of a close friend | 17 | 37 | 8 | 64.05 (1.12) | 61.08, 66.8 | 44.90 | 50 (40–70) | 27.1 | 2 |
| Pregnancy | 12 | 40 | 6 | 64.66 (1.06) | 62.03, 67.27 | 47.63 | 60 (40–70) | 24.7 | 3 |
| Change in residence | 32 | 20 | 19 | 42.69 (0.95) | 40.33, 44.99 | 34.54 | 40 (30–59.8) | 22.7 | 4 |
| Major change in work hours or conditions | 31 | 20 | 27 | 37.09 (0.84) | 34.76, 39.32 | 20.44 | 25 (10–40) | 17.1 | 5 |
| Major change in sleeping habits | 38 | 16 | 30 | 31.92 (0.84) | 29.83, 34.3 | 24.85 | 30 (20–40) | 15.9 | 6 |
| Changing to a new school | 33 | 20 | 28 | 34.6 (0.93) | 32.29, 36.95 | 29.50 | 35 (20–50) | 14.6 | 7 |
| Major change in living conditions | 28 | 25 | 24 | 39.36 (0.9) | 37.01, 41.75 | 23.76 | 30 (20–50) | 14.4 | 8 |
| Spouse/life partner begins or stops working | 26 | 26 | 22 | 40.06 (0.9) | 37.72, 42.36 | 21.22 | 30 (10–50) | 14.1 | 9 |
| Major change in financial state | 16 | 38 | 12 | 52.02 (0.94) | 49.68, 54.68 | 35.47 | 50 (30–65) | 14.0 | 10 |
| Losing your job | 8 | 47 | 10 | 60.97 (1.03) | 58.2, 63.5 | 24.33 | 30 (20–50) | 14.0 | 11 |
| Detention in jail or other institution | 4 | 63 | 2 | 76.88 (1.14) | 73.79, 79.66 | 61.65 | 80 (70–99) | 13.9 | 12 |
| Death of a spouse or life partner | 1 | 100 | 1 | 86.83 (0.98) | 84.16, 89.21 | 73.78 | 95 (80–100) | *-13.2* | *13* |
| Death of a close family member | 5 | 63 | 3 | 75.84 (1.02) | 73.13, 78.46 | 67.57 | 80 (60.5–95) | 12.8 | 14 |
| Gaining a new family member | 14 | 39 | 13 | 51.81 (1.11) | 49.05, 54.65 | 29.09 | 40 (20–50) | 12.8 | 15 |
| Son or daughter leaving home | 23 | 29 | 21 | 41.66 (0.99) | 39.2, 44.19 | 21.75 | 30 (15–40) | 12.7 | 16 |
| Major change in eating habits | 40 | 15 | 35 | 27.39 (0.77) | 25.2, 29.47 | 15.72 | 20 (10–30) | 12.4 | 17 |
| Major change in the health or behaviour of a family member | 11 | 44 | 11 | 55.73 (0.98) | 53.06, 58.24 | 38.01 | 50 (30–70) | 11.7 | 18 |
| Major personal injury or illness | 6 | 53 | 7 | 64.36 (0.98) | 61.9, 66.91 | 55.98 | 70 (50–80) | 11.4 | 19 |
| Taking on a mortgage or loan for a major purchase | 20 | 31 | 20 | 42.22 (0.99) | 39.78, 44.79 | 34.98 | 45 (30–60) | 11.2 | 20 |
| Minor violations of the law | 43 | 11 | 40 | 22.14 (0.76) | 20.12, 24.19 | 14.99 | 20 (10–30) | 11.1 | 21 |
| Major change in usual type and/or amount of recreation | 34 | 19 | 34 | 29.08 (0.78) | 27.01, 31.04 | 11.43 | 15 (5–30) | 10.1 | 22 |
| Major change in the number of arguments with spouse-life partner | 19 | 35 | 18 | 44.21 (0.92) | 41.75, 46.56 | 31.10 | 40 (25.5–50) | 9.2 | 23 |
| Major change in responsibilities at work | 22 | 29 | 26 | 37.8 (0.83) | 35.54, 39.94 | 50.24 | 70 (50–80) | 8.8 | 24 |
| Taking on a loan for a lesser purchase | 37 | 17 | 36 | 25.12 (0.84) | 22.97, 27.6 | 17.36 | 20 (10–35) | 8.1 | 25 |
| Beginning or ceasing formal schooling | 27 | 26 | 29 | 33.88 (0.91) | 31.57, 36.14 | 31.07 | 40 (25–55) | 7.9 | 26 |

*(Continued)*

**Table 3.** (Continued)

| Event | Holmes & Rahe | | Present Study | | | | | Difference Score | |
|---|---|---|---|---|---|---|---|---|---|
| | original rank | weight | present rank | Mean (SE)[a] | 99% CI[b] | Geometric Mean[c] | Median (IQR) | present—original weight | rank: absolute change[d] |
| Major change in number of family get-togethers | 39 | 15 | 38 | 22.88 (0.76) | 20.9, 24.92 | 20.74 | 25 (10–40) | 7.9 | 27 |
| Christmas | 42 | 12 | 43 | 19.78 (0.84) | 17.79, 21.93 | 11.80 | 10 (5–30) | 7.8 | 28 |
| Major business readjustment | 15 | 39 | 16 | 46.73 (1.06) | 43.96, 49.49 | 40.18 | 50 (30.8–70) | 7.7 | 29 |
| Vacation | 41 | 13 | 42 | 20.09 (0.81) | 18.03, 22.25 | 12.64 | 10 (6–30) | 7.1 | 30 |
| Major change in social activities | 36 | 18 | 37 | 24.39 (0.8) | 22.31, 26.45 | 17.35 | 20 (10–30) | 6.4 | 31 |
| Troubles with the boss | 30 | 23 | 33 | 29.15 (0.9) | 26.72, 31.74 | 15.66 | 20 (10–30) | 6.2 | 32 |
| Divorce | 2 | 73 | 4 | 67.86 (1.03) | 65.2, 70.41 | 56.03 | 70 (52.8–85) | *-5.1* | *33* |
| Retirement from work | 10 | 45 | 15 | 49.64 (1.08) | 46.73, 52.45 | 35.58 | 50 (30–60) | 4.6 | 34 |
| Changing to a different line of work | 18 | 36 | 23 | 39.48 (0.85) | 37.3, 41.68 | 52.64 | 70 (45.8–80) | 3.5 | 35 |
| Outstanding personal achievement | 25 | 28 | 32 | 30.94 (0.96) | 28.49, 33.55 | 30.56 | 38.5 (21.3–50) | 2.9 | 36 |
| In-law troubles | 24 | 29 | 31 | 30.94 (0.91) | 28.62, 33.26 | 31.31 | 40 (25–60) | 1.9 | 37 |
| Marital separation | 3 | 65 | 5 | 66.9 (1.01) | 64.28, 69.33 | 55.98 | 70 (50–80) | 1.9 | 38 |
| Marital reconciliation | 9 | 45 | 17 | 46.24 (0.97) | 43.77, 48.59 | 51.32 | 65 (45–80) | 1.2 | 39 |
| Revision of personal habits | 29 | 24 | 39 | 22.8 (0.77) | 20.95, 24.85 | 30.96 | 40 (25–53.8) | *-1.2* | *40* |
| Major change in religious activities | 35 | 19 | 41 | 20.09 (0.8) | 18.01, 22.2 | 22.07 | 30 (15–40) | 1.1 | 41 |
| Sexual difficulties | 13 | 39 | 25 | 38.07 (0.92) | 35.74, 40.61 | 53.44 | 70 (50–80) | *-0.9* | *42* |
| Marriage (pre-set weight) | 7 | 50 | 14 | 50.00 | | | | 0.0 | 43 |
| Single person, living alone | | | | 38.16 (1.13) | 35.41, 41.16 | 24.14 | 40 (15–50) | | |

\* Table's values are ordered by absolute change in weights.

[a] Mean (SE). Standard error obtained via BCa Bootstrap with 1000 samples.

[b] 99% confidence intervals obtained via BCa Bootstrap with 1000 samples.

[c] Where participants responded with a zero value, these were replaced with '1' to allow this value to be calculated.

[d] Rank is based on absolute change (difference score: 2022 weight—original weight). Negative signs were ignored in creating the rank order. Rank values in red indicate that the weight upon which it is based was higher in the original version.

'Major change in social activities' (20). Table 3 shows that the largest BCa confidence interval was for 'Detention in jail or other institution' (73.79–79.66) and the smallest was for 'Revision of personal habits' (20.95–24.85). Of the largest BCa confidence intervals, 3 coincided with some of the largest IQR items: 'gaining a new family member' (49.05–54.65), 'Retirement from work' (46.73–52.45) and 'Single person, living alone' (35.41–41.16). These results suggest that for 79% of items (34/43) participants were consistent in their ratings across items but for the remaining 21% (9/43) respondents were relatively less consistent.

## The SRRS categorised as 'family', 'personal', 'financial' or 'work' life events

Rahe and colleagues delineated their 43 SRRS items in terms of 'family', 'personal', 'work' and 'financial' life events [24,38]. A copy is provided in Appendix 7 (S7 Appendix). The correlations between the original and new weightings were evaluated by these categories, using Kendall's tau. The result revealed strong, positive associations between original and new weights for family (r = 0.818, p < .001), personal (0.638, p < .001) and work (0.905, p = .004) events. However, original and new weights showed no statistically significant association for financial items (p > .4). Bayesian Kendall's tau correlations confirmed these findings with decisive evidence for family and personal categories (BF > 100) and strong evidence for work-related events (BF = 12.24). For financial items, the Bayesian Kendall's tau revealed anecdotal evidence (BF = 0.68). Thus, original and new samples co-varied on all categories except financial events for which evidence was inconclusive.

Demographic differences were examined within each of the 4 categories. Table 4 provides medians with interquartile ranges for all variables analysed. Appendix 8 (S8 Appendix) provides medians and interquartile ranges for all broad categories for demographical variables.

For age, ratings of young, middle-aged and older-aged groups were compared for each of the 4 categories with Kruskal-Wallis tests, which revealed no statistically significant differences between groups (p's > .2). Bayesian Mann-Whitney U tests were used to compare the respective groups as there is no corresponding Bayesian non-parametric one-way ANOVA equivalent. The outcomes were congruent with the frequentist findings and consistently supported the null (BFs ≤ 0.26). Sensitivity analyses are provided in Appendix 6 (S6 Appendix) and support these BF results. Comparing female and male average ratings, in contrast, revealed a statistically significant difference using Mann-Whitney U tests for family events (Mdn 57.1 vs. 50.4), personal events (Mdn 36.9 vs. 32.3), financial events (Mdn 47.5 vs. 42.5) and work events (Mdn 45.7 vs. 40) (p's < .001) with females' ratings being consistently higher than males', respectively. Bayesian Mann-Whitney U tests revealed strong evidence for financial items (BF = 27.22) and decisive evidence for all other categories (BF > 100). Sensitivity analyses are provided in Appendix 6 (S6 Appendix) and support these BFs. These results are shown in Fig 1. Ethnicity, religion, relationship status and employment variables were collapsed into dichotomised variables to simplify comparison. Details are given in Table 4. Appendix 8 (S8 Appendix) provides comparisons for full variables. For ethnicity, a Mann-Whitney U test of white vs. (combined) non-white sub-sets indicated no statistically significant between-groups differences for any of the 4 categories (p's > .1). Likewise, the Bayesian analyses revealed evidence for the null for all comparisons (BFs ≤ 0.20). For religion, a Mann-Whitney U test comparing no-religion vs. (combined) religion groups revealed a statistically significant difference for personal events (Z = -2.006, p = .047) with the no-religion group assigning a higher average weight to this category of events than the religion group (Mdn 35.7 vs. 33.9, respectively). However, the Bayesian Mann-Whitney U test revealed anecdotal evidence (BF = 0.59). Comparisons for family, work and financial categories were not statistically significant (p's > .1) which was confirmed by the Bayesian results which supported the null (BFs ≤ 0.28). For relationship status, a Mann-Whitney U test comparing (combined) married vs. (combined) unmarried groups showed a statistically significant difference for family events (Z = -2.144, p = .032) only with the married group giving higher ratings than the unmarried group (Mdn 55 vs. 53.6, respectively). However, the Bayesian Mann-Whitney U test revealed anecdotal evidence (BF = 1.74). The comparisons for personal, financial and work were not statistically significant (p's > .3), confirmed by Bayesian evidence for the null (BFs ≤ 0.16). For employment status, a Mann-Whitney U test comparing employed vs. (combined) unemployed groups revealed no statistically significant differences (p's > .2) for any of the categories. Congruent

**Table 4. Descriptive statistics for SRRS events categorised by family, financial, personal and work.**

| | sub-groups | overall weight | Mean SRRS weights | | | |
|---|---|---|---|---|---|---|
| | | | family items | financial items | personal items | work items |
| Age | | | Median (IQR) | Median (IQR) | Median (IQR) | Median (IQR) |
| | < 30 years (n = 116) | 40.3 (30.3–54.3) | 53.6 (44.8–61) | 50 (33.8–55.8) | 35.5 (28.6–45.3) | 42.1 (31.6–54.3) |
| | 30 to 60 years (n = 291) | 39.7 (29.7–57.8) | 54.6 (45.7–64.3) | 45 (35–52.5) | 34.7 (28.9–43.3) | 43.6 (32.1–52.9) |
| | > 60 years (n = 133) | 39.4 (28.7–57.1) | 53.9 (46.1–61.1) | 46.5 (35.6–58.8) | 33.4 (26.4–46.6) | 42.9 (33.2–54.7) |
| Sex | | | | | | |
| | female (n = 308) | 43.9 (32.4–58.9) | 57.1 (48.9–64.9) | 47.5 (37.5–56.2) | 36.9 (30.6–46.5) | 45.7 (35.7–55.7) |
| | male (n = 230) | 35.4 (25.3–50.8) | 50.4 (42.4–58.8) | 42.5 (30–55) | 32.3 (24.7–40.5) | 40 (29.3–50) |
| Ethnicity | | | | | | |
| | white (n = 455) | 39.3 (28.7–56.3) | 54.3 (46.4–61.8) | 46.3 (35–55) | 34.2 (27.9–43.7) | 42.9 (32.9–52.9) |
| | non-white[a] (n = 85) | 40.7 (32.5–53) | 55.4 (44.8–63.8) | 50 (37.4–59.4) | 37.9 (27.4–47.9) | 45.7 (31.8–55.7) |
| Religion | | | | | | |
| | no religion (n = 274) | 39 (28.1–55.4) | 55.5 (46.4–64.3) | 47.5 (35–57.5) | 35.7 (27.7–46.6) | 44.3 (33.4–54.4) |
| | religion[b] (n = 266) | 40.9 (30.1–56) | 53.6 (45.6–60.7) | 45 (35–54.1) | 33.9 (28–41.1) | 42.1 (32.1–52.1) |
| Relationship status | | | | | | |
| | married[c] (n = 343) | 40.7 (30.6–57.6) | 55 (46.8–64.5) | 46.5 (35–55) | 34.7 (28.4–45.3) | 42.9 (32.9–54.3) |
| | unmarried (n = 197) | 37.7 (29.8–53.2) | 53.6 (44.6–60.7) | 45.8 (34.8–55.6) | 34.7 (27.1–44.2) | 43.6 (32.5–52.9) |
| Employment status | | | | | | |
| | employed (n = 348) | 39.8 (28.3–55.9) | 54.3 (46.1–63) | 46 (35–55) | 34.7 (28.2–44.5) | 42.9 (32.9–55) |
| | unemployed[e] (n = 192) | 39.2 (30.7–55.4) | 54.3 (45.8–61.1) | 47 (35–57.5) | 34.7 (27.3–44.7) | 43.2 (31.6–52.6) |

[a] 'Non-white' included all ethnicities: mixed race, Asian (southern/southeastern Asia), Chinese (east Asian), black (any region).

[b] 'Religious' includes all religions: Christian, Buddhist, Hindu, Jewish, Muslim, Sikh, any other religion.

[c] 'married' includes married and long-term partners.

[d] 'unmarried' includes those who don't qualify as 'c': divorced, in a relationship, life partner died, separated, single, widowed.

[e] currently unemployed and looking for work, have never worked, long-term sick/disabled, looking after home or family, retired, student (p/t or f/t) and currently unemployed, other.

with this outcome, Bayesian Mann-Whitney U tests revealed evidence for the null for all comparisons (BFs ≤ 0.20). Sensitivity analyses are provided in Appendix 6 (S6 Appendix) for all the above-mentioned Bayesian Mann-Whitney U comparisons and support the reported BFs.

## SRRS weights: Comparing normative and 70+ sub-samples

Muhlenkamp and colleagues [12] who extended the original SRRS by adding weights to represent those aged ≥ 65 to 84 years compared the original 1967 normative sample's ratings (<30 years to > 60 years, n = 394) with those from their new, older group (65 to 84 yrs, n = 41). A similar approach was followed here, however the present study's older group was 70 to 84 years to minimise overlap with previously represented older age groups (e.g. 65 to 69 year-olds). The original study stated that there were 51 raters > 60 years (i.e. no maximum age was reported) while Muhlenkamp and colleagues [12] stated in their report that the original SRRS did not include adults over 70 years. Thus, in the present study, the sample was grouped as those aged 18 to 69 years ('normative sample', n = 473) vs. 70+ years ('70+ sample', n = 67). Between-groups differences were evaluated overall as well as within the previously mentioned 4 life events categories: 'family', 'personal', 'work' and 'financial'.

In the overall assessment the normative sample's summed total LCUs were higher ($LCU_{total}$ 1829) than that of the 70+ group ($LCU_{total}$ 1759), however a Mann-Whitney U test found no statistically significant difference between the two groups, on average (p > .7). Likewise, the

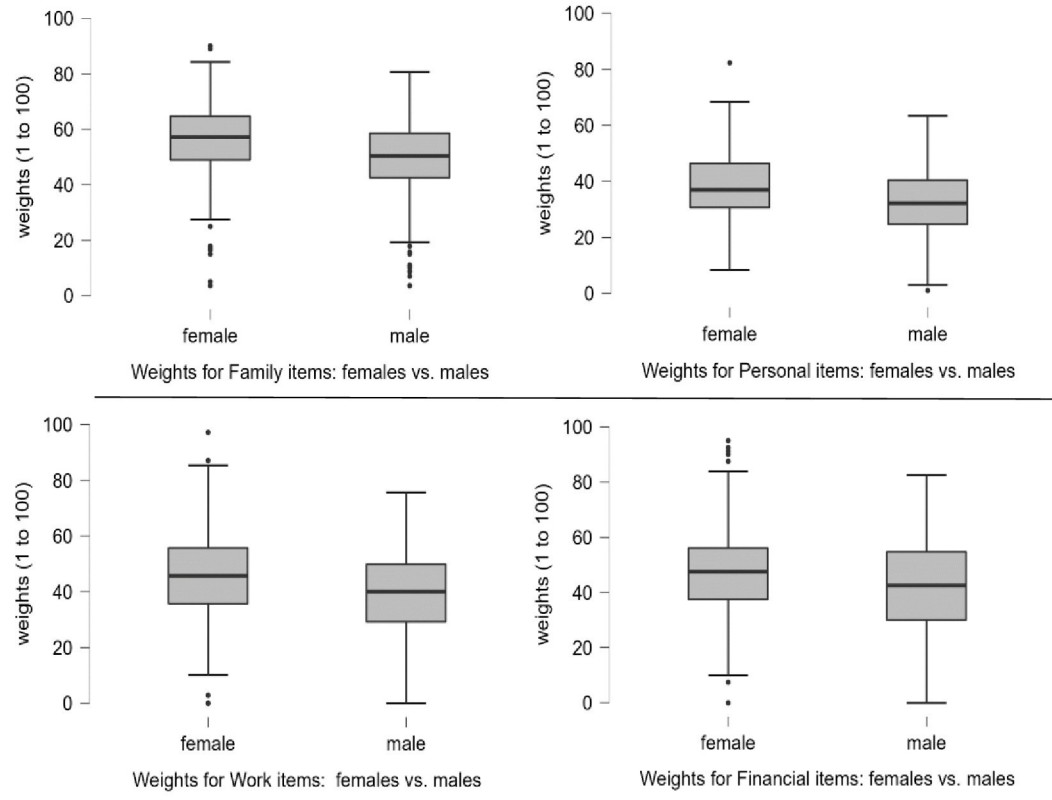

**Fig 1. Box plots showing differences between males and females based on event categories.** From top-left to bottom-right the box plots shown are: 'family', 'personal', 'work' and 'financial'.

Bayesian equivalent supported the null (BF = 0.24). Sensitivity analyses (S6 Appendix) were congruent with this outcome. Further, the Kendall's tau indicated that the lists were strongly, positively correlated (r = .884, p < .001). The corresponding Bayesian Kendall's tau provided decisive evidence for this finding (BF > 100). Of the 42 original items rated, only 12 were higher for the 70+ group. Thus, the normative and older samples co-varied strongly regarding the ratings, though the normative sample's ratings were consistently higher for most items. To assess whether there were any systematic differences in ratings between the normative group and 70+ group based on the 4 categories, a chi-square was conducted with event category (family, personal, financial, work) and proportion of change (> adjustment required in 70 + participants vs. > adjustment required in normative participants). The association was not statistically significant (p = .648). Likewise, the Bayesian contingency tables test supported the null (BF = 0.17). These findings suggest that there were no significant age-based differences in ratings across the different categories of events.

## SRRS weights: Comparing young, middle-aged and older adults

A set of SRRS weights and ranks for each age group were created and are provided in Table 5. Summing the weights by age group, it was found that YAs' summed weights value or total life change units (LCU$_{total}$) was 1866, for MAs the LCU$_{total}$ was 1875 and for OAs it was 1867. A Kruskal-Wallis test revealed no statistically significant differences (p > .997). Bayesian non-parametric Mann-Whitney U comparisons agreed with these findings, showing evidence for the null (BFs = 0.23). Bayes factor sensitivity analyses (S6 Appendix) were comparable. These

**Table 5.** SRRS events by young, middle-aged and older adults' weights and ranks.

| Event | Holmes & Rahe original rank | Present study Young adults rank | Mean (SE)[a] | 99% CI[b] | Middle-aged adults rank | Mean (SE)[a] | 99% CI[b] | Older adults rank | Mean (SE)[a] | 99% CI[b] |
|---|---|---|---|---|---|---|---|---|---|---|
| Death of a spouse or life partner | 1 | 1 | 85.65 (2.1) | 79.53, 90.65 | 1 | 87.72 (1.24) | 84.24, 90.8 | 1 | 85.92 (2.07) | 80.27, 91.08 |
| Divorce | 2 | 6 | 65.13 (2.15) | 59.34, 70.87 | 4 | 68.28 (1.39) | 64.66, 71.64 | 4 | 69.34 (2.05) | 63.45, 74.43 |
| Marital separation | 3 | 7 | 62.17 (2.15) | 56.95, 67.54 | 5 | 68.09 (1.33) | 64.49, 71.59 | 5 | 68.44 (1.92) | 63.1, 73.18 |
| Detention in jail or other institution | 4 | 3 | 70.65 (2.53) | 63.62, 76.69 | 2 | 79.74 (1.38) | 76.4, 83.08 | 2 | 76.08 (2.4) | 69.22, 81.99 |
| Death of a close family member | 5 | 2 | 75.26 (2.44) | 68, 80.98 | 3 | 76.92 (1.38) | 73.26, 80.32 | 3 | 74 (2.02) | 68.54, 79.04 |
| Major personal injury or illness | 6 | 8 | 59.84 (2.14) | 53.53, 65.63 | 6 | 65.72 (1.31) | 62.46, 68.98 | 7 | 65.33 (1.89) | 60.53, 70.24 |
| Marriage (pre-set weight) | 7 | 14 | 50 | | 14 | 50 | | 13 | 50 | |
| Losing your job | 8 | 9 | 59.05 (2.22) | 52.93, 64.54 | 10 | 61.7 (1.37) | 57.99, 65.51 | 9 | 61.06 (2.2) | 55.08, 67.03 |
| Marital reconciliation | 9 | 18 | 44.11 (1.95) | 38.77, 49.37 | 16 | 47.01 (1.36) | 43.26, 50.26 | 17 | 46.41 (1.9) | 41.07, 51.2 |
| Retirement from work | 10 | 13 | 50.16 (2.38) | 44.24, 56.17 | 15 | 49.9 (1.47) | 46.34, 53.55 | 16 | 48.63 (2.18) | 42.8, 53.99 |
| Major change in the health or behaviour of a family member | 11 | 15 | 49.13 (2.08) | 43.87, 54.63 | 11 | 57.75 (1.31) | 54.44, 61.2 | 11 | 57.06 (1.9) | 51.71, 62.19 |
| Pregnancy | 12 | 5 | 66.61 (2.45) | 60.49, 72.75 | 8 | 64.19 (1.36) | 60.74, 67.54 | 8 | 63.97 (1.99) | 58.47, 68.63 |
| Sexual difficulties | 13 | 30 | 34.47 (1.9) | 29.8, 39.8 | 25 | 38.94 (1.28) | 35.54, 42.3 | 24 | 39.29 (1.91) | 33.98, 44.56 |
| Gaining a new family member | 14 | 12 | 52.8 (2.38) | 47.02, 59.34 | 12 | 52.52 (1.55) | 48.36, 56.6 | 15 | 49.41 (2.02) | 44.34, 54.25 |
| Major business readjustment | 15 | 17 | 44.69 (2.27) | 37.94, 50.73 | 17 | 46.19 (1.43) | 42.65, 49.8 | 14 | 49.68 (2.24) | 44.15, 55.21 |
| Major change in financial state | 16 | 11 | 54.3 (2.04) | 49.06, 59.65 | 13 | 51.59 (1.24) | 48.35, 54.94 | 12 | 50.96 (1.95) | 45.2, 56.64 |
| Death of a close friend | 17 | 4 | 69.68 (2.37) | 63.06, 75.74 | 7 | 64.99 (1.54) | 60.83, 68.64 | 10 | 57.09 (2.17) | 51.23, 62.53 |
| Changing to a different line of work | 18 | 22 | 40.88 (2) | 35.67, 45.97 | 23 | 39.66 (1.14) | 36.82, 42.83 | 26 | 37.86 (1.83) | 33.26, 42.16 |
| Major change in the number of arguments with spouse/life partner | 19 | 19 | 42.55 (1.91) | 37.52, 47.67 | 18 | 44.42 (1.27) | 41.01, 47.6 | 19 | 45.22 (2.01) | 39.13, 50.98 |
| Taking on a mortgage or loan for a major purchase | 20 | 16 | 46.09 (2.25) | 40.23, 52.17 | 22 | 40 (1.28) | 37.09, 44.12 | 20 | 43.71 (2) | 38.3, 48.95 |
| Foreclosure/repossession on mortgage or loan | 21 | 10 | 55.22 (2.23) | 49.49, 61.59 | 9 | 63.09 (1.41) | 59.57, 66.72 | 6 | 67.11 (2.27) | 60.81, 72.56 |
| Major change in responsibilities at work | 22 | 26 | 38.16 (1.79) | 33.27, 42.88 | 28 | 37.09 (1.1) | 34.21, 40.3 | 25 | 39.02 (1.73) | 34.25, 43.78 |
| Son or daughter leaving home | 23 | 21 | 40.92 (1.97) | 35.67, 45.87 | 19 | 41.98 (1.38) | 38.62, 45.75 | 22 | 41.61 (1.99) | 36.39, 46.26 |
| In-law troubles | 24 | 34 | 30.3 (1.99) | 25.21, 35.09 | 32 | 30.25 (1.18) | 27.05, 33.34 | 31 | 33.02 (1.85) | 28.48, 38.37 |
| Outstanding personal achievement | 25 | 36 | 29.21 (2.07) | 24.57, 34.16 | 34 | 29.66 (1.23) | 26.45, 33.05 | 28 | 35.26 (1.98) | 30.36, 40.52 |

*(Continued)*

**Table 5.** (Continued)

| Event | Holmes & Rahe | Present study | | | | | | | | |
|---|---|---|---|---|---|---|---|---|---|---|
| | | Young adults | | | Middle-aged adults | | | Older adults | | |
| | original rank | rank | Mean (SE)[a] | 99% CI[b] | rank | Mean (SE)[a] | 99% CI[b] | rank | Mean (SE)[a] | 99% CI[b] |
| Spouse/life partner begins or stops working | 26 | 23 | 40.34 (1.89) | 35.55, 45.12 | 20 | 41.88 (1.25) | 38.85, 45.73 | 27 | 35.84 (1.85) | 31.21, 40.86 |
| Beginning or ceasing formal schooling | 27 | 28 | 36.34 (2) | 30.93, 41.56 | 31 | 33.07 (1.18) | 30.26, 36.41 | 30 | 33.51 (2.02) | 28.31, 39.14 |
| Major change in living conditions | 28 | 24 | 39.28 (2.03) | 34.28, 45.03 | 24 | 39.38 (1.24) | 36.05, 42.25 | 23 | 39.38 (1.87) | 34.36, 44.22 |
| Revision of personal habits | 29 | 38 | 27.25 (1.93) | 22.7, 32.39 | 41 | 21.56 (1) | 19.19, 24.34 | 42 | 21.65 (1.39) | 17.89, 25.55 |
| Troubles with the boss | 30 | 40 | 24.69 (1.84) | 20.51, 29.09 | 33 | 30.03 (1.25) | 26.94, 33.46 | 33 | 31.12 (2.01) | 25.7, 36.14 |
| Major change in work hours or conditions | 31 | 25 | 38.64 (1.8) | 33.95, 43.11 | 26 | 38.41 (1.15) | 35.22, 42.12 | 32 | 32.85 (1.69) | 28.43, 37.43 |
| Change in residence | 32 | 20 | 41.01 (2.04) | 36.02, 46.2 | 21 | 41.75 (1.29) | 38.69, 44.92 | 18 | 46.23 (1.81) | 41.48, 50.53 |
| Changing to a new school | 33 | 27 | 37.5 (2.02) | 32.94, 42.68 | 29 | 33.51 (1.25) | 30.37, 36.51 | 29 | 34.44 (2.12) | 28.5, 40.14 |
| Major change in usual type and/or amount of recreation | 34 | 31 | 33.24 (1.86) | 28.83, 38.04 | 35 | 28.65 (1.02) | 26.2, 31.28 | 35 | 26.38 (1.5) | 22.8, 30.77 |
| Major change in religious activities | 35 | 39 | 26.08 (1.7) | 22.08, 30.2 | 44 | 18.05 (1.02) | 15.55, 20.85 | 44 | 19.33 (1.7) | 14.88, 24.21 |
| Major change in social activities | 36 | 37 | 28.78 (1.9) | 24.2, 33.82 | 38 | 23.3 (1.06) | 20.31, 26.5 | 39 | 22.95 (1.4) | 19.4, 26.91 |
| Taking on a loan for a lesser purchase | 37 | 35 | 29.83 (1.95) | 25.3, 35.2 | 37 | 23.51 (1.1) | 20.85, 26.6 | 36 | 24.55 (1.66) | 20.33, 28.88 |
| Major change in sleeping habits | 38 | 32 | 32.26 (2.13) | 27.13, 37.73 | 30 | 33.25 (1.12) | 30.21, 36.13 | 34 | 28.74 (1.49) | 25.24, 32.57 |
| Major change in number of family get-togethers | 39 | 41 | 24.53 (1.88) | 20.13, 29.1 | 40 | 21.73 (0.94) | 19.44, 24.4 | 38 | 23.97 (1.56) | 20.28, 28.06 |
| Major change in eating habits | 40 | 33 | 30.68 (1.94) | 26.24, 36.11 | 36 | 27.63 (1.06) | 24.81, 30.71 | 37 | 24 (1.36) | 20.63, 28.16 |
| Vacation | 41 | 42 | 21.45 (2.05) | 16.28, 26.5 | 43 | 19.13 (1.07) | 16.33, 22.04 | 43 | 21 (1.51) | 17.29, 25.52 |
| Christmas | 42 | 44 | 16.95 (1.7) | 12.65, 21.35 | 42 | 19.79 (1.09) | 16.76, 22.86 | 41 | 22.23 (1.93) | 17.62, 27.65 |
| Minor violations of the law | 43 | 43 | 19.85 (1.47) | 16.66, 24.09 | 39 | 22.72 (1.04) | 20, 25.45 | 40 | 22.87 (1.64) | 18.4, 27.3 |
| Single person, living alone | | 29 | 34.96 (2.59) | 28.55, 41.39 | 27 | 37.84 (1.49) | 33.92, 41.51 | 21 | 41.66 (2.28) | 35.82, 48.26 |

[a] SE = Standard Error.

[b] CI = Confidence Intervals.

results indicated comparable overall weights across the life span. In assessing the strength of association across all items between the pairs of age groups (YA vs. MA; MA vs. OA; YA vs. OA), Kendall's tau correlation coefficients indicated very strong, positive correlations (r's ≥ .835, p's < .001). Bayesian Kendall's tau coefficients agreed with these findings (BFs > 100).

## SRRS weights: Comparing males and females

Miller and Rahe (1997) found that females' ratings were 17% higher on average than that of males. Table 6 provides the descriptive statistics for the present samples' weights and ranking by sex. Females' summed weights (LCU$_{total}$ 1992) were found to be 14% higher than those for males (LCU$_{total}$ 1708). The corresponding Mann-Whitney U test revealed a trend (z = -1.840, p = .066, r = .2), suggesting that the average difference between males' (Mdn 35.3, n = 43) and females' (Mdn 45.1, n = 43) LCU$_{total}$ was statistically comparable. The corresponding Bayesian test revealed anecdotal evidence for this finding (BF = 1.42). Bayes factor sensitivity analyses (S6 Appendix) were comparable. A Kendall's tau correlation coefficient indicated a very strong, positive correlation between males' and females' ratings (r = .892, p < .001), as indicated by Fig 2. The corresponding Bayesian Kendall's tau correlation revealed decisive evidence for this finding (BF > 100). These results suggest that whilst females' ratings were higher than males' on all items, there was strong covariance between them. The 3 items with the greatest difference in ratings, with males' ratings being lower in each case, were 'Death of a close friend' (-15 LCU), 'Spouse/life partner begins or stops working' (-13 LCU) and 'Son or daughter leaving home' (-12 LCU). A likewise comparison regarding the 3 items with the smallest difference were 'Gaining a new family member' (-1 LCU), 'Retirement from work' (-2 LCU) and 'Marital reconciliation' (-3 LCU).

## Impact of personal experience on SRRS ratings

To ascertain whether there was a link between ratings for each life event and personal experience of that item a series of correlations using Kendall's tau were conducted. Table 7 provides the descriptive statistics for personal experience. Of 42 items, 18 showed a statistically significant correlation (p's ≤ .042). However, all coefficients were very small (r ≤ .146). The Bayesian Kendall's tau similarly found evidence ranging from substantial to decisive (BFs ≥ 3.0) for 15 items. Of these, 6 items were supported by ≥ very strong evidence but correlation coefficients remained small with magnitudes ranging from 0.108 to 0.146 as shown in Appendix 9 (S9 Appendix). These results suggest that there may have been some events for which personal experience were weakly, positively associated with event ratings. Overall, however, participants' personal experiences did not appear to systematically bias their ratings.

## Impact of loneliness on SRRS ratings

Overall, participants' average loneliness score (loneliness level), as measured by the R-UCLA (higher value = higher level of loneliness), was M 5.1 (SE .08), ranging from 3 to 9. In response to how often respondents felt lonely (loneliness frequency), with a lower value indicating feeling lonely more often, 9.8% (n = 53) were lonely 'often/always', 24.1% (n = 130) 'some of the time, 24.3% (n = 131) 'occasionally', 27.6% (n = 149) 'hardly ever' while 13% (n = 70) indicated 'never'.

   To explore whether loneliness affected SRRS ratings, Kendall's tau correlational analyses were conducted to test the association between the R-UCLA and loneliness frequency measures and each of the 43 SRRS rating items. The frequentist analyses for both level and frequency of loneliness revealed statistically significant correlations for 7 items (p's ≤ .039), however the correlation coefficients were very small (r ≤ .128). For both level and frequency, 6 items were 'Revision of personal habits'; 'Foreclosure/repossession on mortgage or loan'; 'Detention in jail or other institution'; 'Major change in usual type and/or amount of recreation'; 'Major change in work hours or conditions' and 'Major change in religious activities'. For loneliness level only, 'Major change in sleeping habits' was. For loneliness frequency only 'Major change in social activities' was significant. Bayesian analysis was only conducted for loneliness level as there was no equivalent non-parametric Bayesian analysis for loneliness

**Table 6. SRRS events by females' and males' weights and ranks.**

| Event | Female | | | Male | | |
|---|---|---|---|---|---|---|
| | rank | M (SE) | 99% CI | rank | M (SE) | 99% CI |
| Death of a spouse or life partner | 1 | 89.73 (1.03) | 86.48, 92.62 | 1 | 82.96 (1.75) | 78.09, 87.37 |
| Divorce | 5 | 69.91 (1.31) | 66.38, 73.15 | 4 | 65.1 (1.68) | 61.02, 69.41 |
| Marital separation | 6 | 69.09 (1.23) | 65.79, 72.11 | 5 | 64.34 (1.68) | 59.93, 68.35 |
| Detention in jail or other institution | 3 | 79.28 (1.38) | 75.21, 83.25 | 2 | 73.78 (1.77) | 69.04, 78.63 |
| Death of a close family member | 2 | 80.85 (1.22) | 77.52, 84.07 | 3 | 69.23 (1.76) | 64.72, 74.2 |
| Major personal injury or illness | 8 | 67.33 (1.21) | 64.07, 70.62 | 7 | 60.52 (1.57) | 56.38, 64.28 |
| Marriage (arbitrary weight) | 16 | 50 | | 15 | 50 | |
| Losing your job | 10 | 63.43 (1.3) | 60.34, 66.66 | 9 | 58.09 (1.66) | 53.8, 61.99 |
| Marital reconciliation | 17 | 47.5 (1.16) | 44.07, 51.14 | 16 | 44.83 (1.52) | 41.11, 48.59 |
| Retirement from work | 15 | 50.39 (1.37) | 46.91, 54.32 | 14 | 48.51 (1.7) | 44.07, 52.58 |
| Major change in the health or behaviour of a family member | 11 | 60.28 (1.15) | 56.98, 63.32 | 10 | 49.82 (1.64) | 45.73, 53.98 |
| Pregnancy | 7 | 67.81 (1.35) | 64.44, 71.52 | 6 | 60.78 (1.73) | 56.34, 65.43 |
| Sexual difficulties | 27 | 39.24 (1.17) | 35.88, 42.18 | 26 | 36.52 (1.47) | 32.68, 40 |
| Gaining a new family member | 13 | 52.21 (1.44) | 48.95, 55.73 | 12 | 51.09 (1.8) | 45.9, 55.72 |
| Major business readjustment | 14 | 51.84 (1.33) | 48.6, 55.11 | 13 | 39.73 (1.61) | 35.45, 44.14 |
| Major change in financial state | 12 | 54.58 (1.21) | 51.52, 57.64 | 11 | 48.44 (1.49) | 44.56, 52.34 |
| Death of a close friend | 4 | 70.24 (1.36) | 66.8, 73.76 | 10 | 55.62 (1.88) | 51.5, 60.37 |
| Changing to a different line of work | 24 | 42.67 (1.16) | 39.52, 45.87 | 23 | 35.33 (1.26) | 32.05, 38.56 |
| Major change in the number of arguments with spouse-life partner | 18 | 47.12 (1.23) | 43.79, 50.4 | 17 | 40.27 (1.42) | 36.54, 43.59 |
| Taking on a mortgage or loan for a major purchase | 22 | 45.12 (1.25) | 42.06, 48.23 | 21 | 38.09 (1.49) | 33.95, 41.64 |
| Foreclosure/repossession on mortgage or loan | 9 | 65.23 (1.44) | 61.56, 68.54 | 8 | 58.65 (1.75) | 54.29, 63.21 |
| Major change in responsibilities at work | 26 | 40.72 (1.13) | 37.81, 43.77 | 25 | 33.86 (1.23) | 30.51, 37.15 |
| Son or daughter leaving home | 19 | 47.02 (1.27) | 44.07, 50.45 | 18 | 34.71 (1.45) | 30.96, 38.58 |
| In-law troubles | 30 | 35.04 (1.18) | 32.08, 38.33 | 29 | 25.56 (1.27) | 22.25, 28.63 |
| Outstanding personal achievement | 32 | 33.43 (1.35) | 30.1, 36.98 | 31 | 27.58 (1.31) | 23.73, 31.23 |
| Spouse/life partner begins or stops working | 20 | 45.73 (1.17) | 42.81, 48.54 | 19 | 32.25 (1.24) | 29.04, 35.63 |
| Beginning or ceasing formal schooling | 29 | 36.79 (1.2) | 33.98, 40.25 | 28 | 30.07 (1.38) | 26.45, 33.54 |
| Major change in living conditions | 23 | 42.76 (1.2) | 39.5, 46.39 | 22 | 34.89 (1.4) | 31, 38.96 |
| Revision of personal habits | 39 | 25.59 (1.11) | 22.56, 28.84 | 38 | 18.97 (1.03) | 16.52, 21.82 |
| Troubles with the boss | 33 | 32.07 (1.26) | 28.61, 35.55 | 32 | 25.19 (1.3) | 21.83, 28.38 |
| Major change in work hours or conditions | 25 | 41.06 (1.11) | 37.97, 44.21 | 24 | 31.7 (1.23) | 28.6, 34.82 |
| Change in residence | 21 | 45.63 (1.22) | 42.38, 48.8 | 20 | 38.79 (1.5) | 34.66, 42.62 |
| Changing to a new school | 28 | 38.2 (1.29) | 34.82, 41.54 | 27 | 29.76 (1.34) | 26.23, 33.14 |
| Major change in usual type and/or amount of recreation | 34 | 31.81 (1.03) | 29.2, 34.92 | 33 | 25.28 (1.11) | 22.33, 28.14 |
| Major change in religious activities | 42 | 21.8 (1.08) | 19.25, 24.56 | 41 | 17.77 (1.2) | 14.86, 21.02 |
| Major change in social activities | 36 | 26.93 (1.03) | 24.45, 29.45 | 35 | 20.73 (1.12) | 17.94, 23.51 |
| Taking on a loan for a lesser purchase | 37 | 26.53 (1.12) | 23.56, 29.65 | 36 | 23.19 (1.24) | 19.64, 26.52 |
| Major change in sleeping habits | 31 | 34.81 (1.15) | 31.64, 37.69 | 30 | 27.77 (1.22) | 24.9, 31.18 |
| Major change in number of family get-togethers | 38 | 25.93 (1.08) | 23.17, 28.91 | 37 | 18.78 (1) | 16.27, 21.23 |
| Major change in eating habits | 35 | 29.11 (1.03) | 26.46, 31.62 | 34 | 24.89 (1.15) | 22.09, 27.91 |
| Vacation | 41 | 22.01 (1.12) | 19.34, 24.97 | 40 | 17.21 (1.1) | 14.33, 20.1 |
| Christmas | 43 | 21.66 (1.16) | 18.87, 24.53 | 42 | 17.38 (1.18) | 14.5, 20.74 |
| Minor violations of the law | 40 | 23.82 (1.06) | 21.32, 26.44 | 39 | 19.87 (1.14) | 17.18, 22.73 |
| Single person, living alone* | | 40.10 (1.55) | 35.91, 44.52 | | 35.37 (1.76) | 30.48, 39.73 |

*'Single person, living alone' was not included in the ranking as it was not included in the original rating of the SRRS.

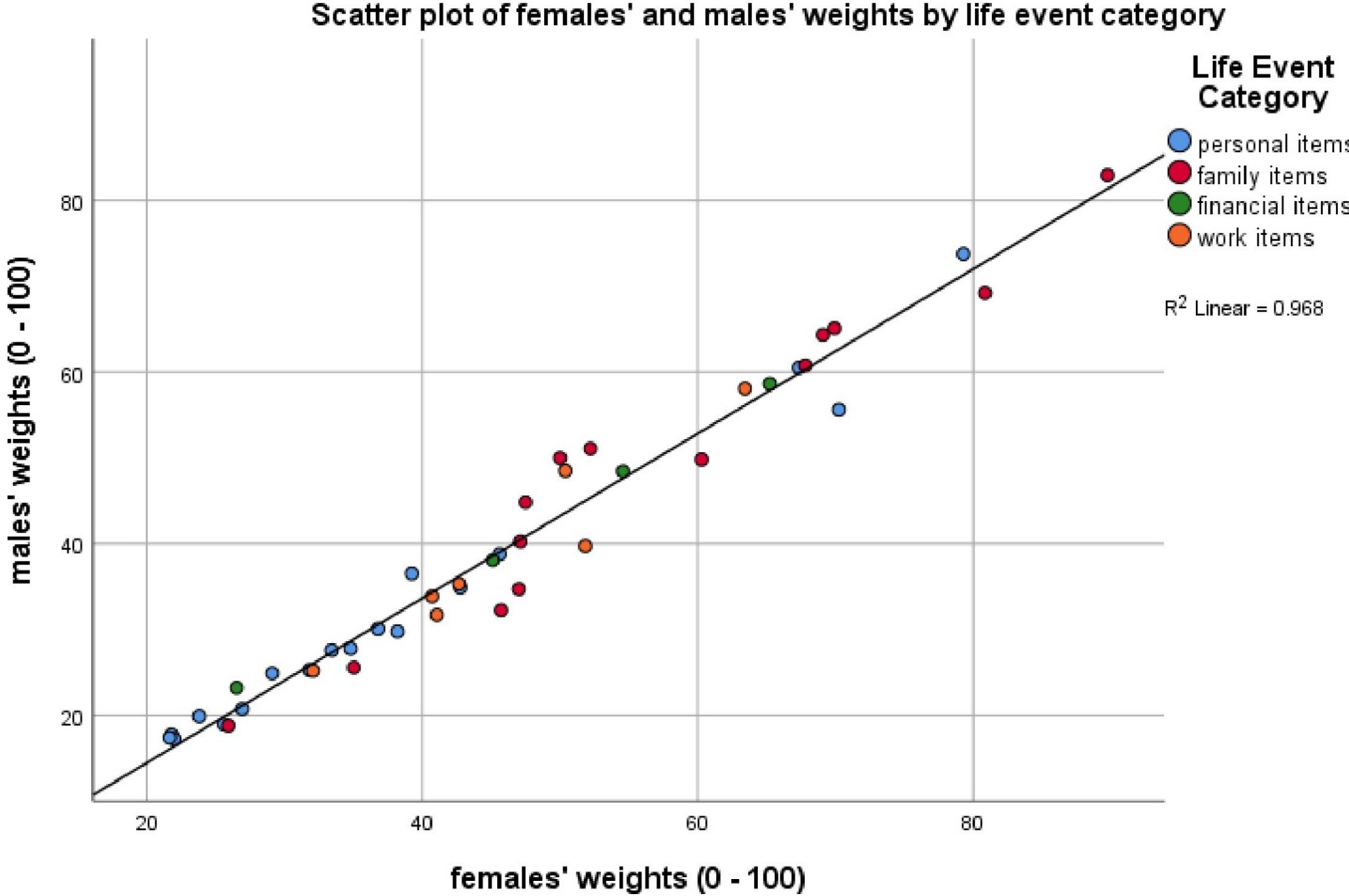

**Fig 2. Scatter plot showing the covariance of females' and males' weights by items.** Scatter plot shows the covariance of females' and males' weights by items with weights (life change units) ranging from 0 to 100, grouped by category: 'family', 'financial', 'personal' and 'work'.

frequency in JASP. The Bayesian Kendall's tau conducted between R-UCLA scores and ratings revealed only two noteworthy results: decisive evidence for a small positive correlation (r = 0.128, BF > 100) between 'Revision of personal habits' and R-UCLA scores and substantial evidence for 'Single person, living alone' and R-UCLA scores (r = 0.082, BF = 3.18). These outcomes indicated that an increase in level of loneliness experienced was associated with an increase in the respective ratings. All other associations either supported the null ($n_{ratings}$ = 32; BF ≤ 0.30) or evidence was anecdotal ($n_{ratings}$ = 9; ≤ 0.35 BF ≤ 2.42).

### Extending the SRRS: New items

**Proposed new item: 'Single person, living alone'.** Table 8 provides descriptive statistics and rank for this new item relative to the existing 43 life events. As the table shows, the overall averaged weight based on the arithmetic mean was 38 (SE 1.13), which places its rank as lower than marriage. Non-parametric frequentist and Bayesian statistics were used to evaluate whether ratings for this event differed depending on age, sex, ethnicity, relationship status, employment status or religion. Table 9 presents their descriptive statistics. A Kruskal Wallis test revealed no statistically significant differences between age groups (p > .07). Bayesian Mann-Whitney U tests compared all 2-way age group combinations and similarly found

**Table 7. Descriptive statistics showing the extent to which ratings were based on personal experience.**

| Life event | Mean % (SE)* | Median % (IQR) |
|---|---|---|
| Christmas | 83.39 (1.14) | 99 (76–100) |
| Vacation | 81.35 (1.1) | 93 (68–100) |
| Death of a close family member | 73.11 (1.55) | 90.5 (60–100) |
| Change in residence | 72.96 (1.42) | 85 (59–100) |
| Beginning or ceasing formal schooling | 68.05 (1.56) | 80 (45.5–100) |
| Pregnancy | 56.17 (1.95) | 77.5 (0–100) |
| Major change in the health or behaviour of a family member | 65.4 (1.53) | 75 (41–100) |
| Taking on a mortgage or loan for a major purchase | 58.39 (1.81) | 74 (2–100) |
| Gaining a new family member | 58.74 (1.78) | 70.5 (7.5–100) |
| Major change in sleeping habits | 62.54 (1.53) | 70 (29–100) |
| Changing to a new school | 58.56 (1.7) | 70 (13–100) |
| Troubles with the boss | 58.63 (1.63) | 70 (19.5–99) |
| Major change in financial state | 62.59 (1.43) | 70 (39–94.5) |
| Taking on a loan for a lesser purchase | 54.91 (1.8) | 68 (1.5–100) |
| Changing to a different line of work | 56.08 (1.6) | 66 (17–89.5) |
| Major change in number of family get-togethers | 57.12 (1.49) | 63.5 (26–86) |
| Major change in eating habits | 56.3 (1.55) | 63 (20.5–90) |
| Single person, living alone | 51.99 (1.89) | 61.5 (0–100) |
| Major change in responsibilities at work | 53.77 (1.55) | 61 (18–86) |
| Major change in work hours or conditions | 54.15 (1.5) | 61 (21–81) |
| Outstanding personal achievement | 51.31 (1.62) | 56 (13–85) |
| Major change in usual type and/or amount of recreation | 49.05 (1.51) | 55 (12–77) |
| Major personal injury or illness | 49.93 (1.73) | 54.5 (4–95.5) |
| Major change in social activities | 48.51 (1.5) | 53 (13.5–76) |
| Major change in living conditions | 47.19 (1.66) | 51 (1–82) |
| Losing your job | 46.81 (1.81) | 50.5 (0–93) |
| Major change in the number of arguments with spouse/life partner | 43.12 (1.58) | 44 (1–74) |
| Revision of personal habits | 42.1 (1.56) | 41.5 (3–72) |
| Minor violations of the law | 43.8 (1.8) | 35 (0–89) |
| Spouse/life partner begins or stops working | 42.17 (1.81) | 26.5 (0–90) |
| Death of a close friend | 40.33 (1.84) | 17 (0–90) |
| Sexual difficulties | 33.85 (1.64) | 15 (0–69) |
| In-law troubles | 32.57 (1.67) | 11 (0–66) |
| Major change in religious activities | 25.72 (1.51) | 2.5 (0–51) |
| Son or daughter leaving home | 34.66 (1.86) | 1 (0–87) |
| Marital separation | 26.48 (1.7) | 0 (0–55.5) |
| Death of a spouse or life partner | 17.71 (1.48) | 0 (0–11) |
| Divorce | 25.52 (1.71) | 0 (0–53) |
| Marital reconciliation | 15.4 (1.33) | 0 (0–10) |
| Major business readjustment | 20.75 (1.37) | 0 (0–31) |
| Retirement from work | 28.6 (1.75) | 0 (0–61) |
| Foreclosure/repossession on mortgage or loan | 11.04 (1.11) | 0 (0–3) |
| Detention in jail or other institution | 10.58 (1.18) | 0 (0–0) |

*Ordered from highest to lowest mean % personal experience.

A value of 100 indicates that the rating for an item was completely based on personal experience.

A value of 0 indicates that the rating for an item was not at all based on personal experience.

N = 536.

**Table 8. New vs. original weights and rank order of SRRS events, including new item: 'Single person, living alone'.**

| Event | Holmes & Rahe | | Present Study | | | | |
|---|---|---|---|---|---|---|---|
| | rank | weight | rank | Mean (SE)[a] | 99% CI[b] | Geometric Mean[c] | Mdn (IQR)[d] |
| Death of a spouse or life partner | 1 | 100 | 1 | 86.83 (0.98) | 84.16, 89.21 | 73.78 | 95 (80–100) |
| Detention in jail or other institution | 4 | 63 | 2 | 76.88 (1.14) | 73.79, 79.66 | 61.65 | 80 (70–99) |
| Death of a close family member | 5 | 63 | 3 | 75.84 (1.02) | 73.13, 78.46 | 67.57 | 80 (60.5–95) |
| Divorce | 2 | 73 | 4 | 67.86 (1.03) | 65.2, 70.41 | 56.03 | 70 (52.8–85) |
| Marital separation | 3 | 65 | 5 | 66.9 (1.01) | 64.28, 69.33 | 55.98 | 70 (50–80) |
| Pregnancy | 12 | 40 | 6 | 64.66 (1.06) | 62.03, 67.27 | 47.63 | 60 (40–70) |
| Major personal injury or illness | 6 | 53 | 7 | 64.36 (0.98) | 61.9, 66.91 | 55.98 | 70 (50–80) |
| Death of a close friend | 17 | 37 | 8 | 64.05 (1.12) | 61.08, 66.8 | 44.9 | 50 (40–70) |
| Foreclosure/repossession on mortgage or loan | 21 | 30 | 9 | 62.39 (1.07) | 59.65, 65.09 | 32.71 | 40 (25–60) |
| Losing your job | 8 | 47 | 10 | 60.97 (1.03) | 58.2, 63.5 | 24.33 | 30 (20–50) |
| Major change in the health or behaviour of a family member | 11 | 44 | 11 | 55.73 (0.98) | 53.06, 58.24 | 38.01 | 50 (30–70) |
| Major change in financial state | 16 | 38 | 12 | 52.02 (0.94) | 49.68, 54.68 | 35.47 | 50 (30–65) |
| Gaining a new family member | 14 | 39 | 13 | 51.81 (1.11) | 49.05, 54.65 | 29.09 | 40 (20–50) |
| Marriage (pre-set weight) | 7 | 50 | 14 | 50 | | | |
| Retirement from work | 10 | 45 | 15 | 49.64 (1.08) | 46.73, 52.45 | 35.58 | 50 (30–60) |
| Major business readjustment | 15 | 39 | 16 | 46.73 (1.06) | 43.96, 49.49 | 40.18 | 50 (30.8–70) |
| Marital reconciliation | 9 | 45 | 17 | 46.24 (0.97) | 43.77, 48.59 | 51.32 | 65 (45–80) |
| Major change in the number of arguments with spouse-life partner | 19 | 35 | 18 | 44.21 (0.92) | 41.75, 46.56 | 31.1 | 40 (25.5–50) |
| Change in residence | 32 | 20 | 19 | 42.69 (0.95) | 40.33, 44.99 | 34.54 | 40 (30–59.8) |
| Taking on a mortgage or loan for a major purchase | 20 | 31 | 20 | 42.22 (0.99) | 39.78, 44.79 | 34.98 | 45 (30–60) |
| Son or daughter leaving home | 23 | 29 | 21 | 41.66 (0.99) | 39.2, 44.19 | 21.75 | 30 (15–40) |
| Spouse/life partner begins or stops working | 26 | 26 | 22 | 40.06 (0.9) | 37.72, 42.36 | 21.22 | 30 (10–50) |
| Changing to a different line of work | 18 | 36 | 23 | 39.48 (0.85) | 37.3, 41.68 | 52.64 | 70 (45.8–80) |
| Major change in living conditions | 28 | 25 | 24 | 39.36 (0.9) | 37.01, 41.75 | 23.76 | 30 (20–50) |
| Single person, living alone | | | 25 | 38.16 (1.13) | 35.41, 41.16 | 24.14 | 40 (15–50) |
| Sexual difficulties | 13 | 39 | 26 | 38.07 (0.92) | 35.74, 40.61 | 53.44 | 70 (50–80) |
| Major change in responsibilities at work | 22 | 29 | 27 | 37.8 (0.83) | 35.54, 39.94 | 50.24 | 70 (50–80) |
| Major change in work hours or conditions | 31 | 20 | 28 | 37.09 (0.84) | 34.76, 39.32 | 20.44 | 25 (10–40) |
| Changing to a new school | 33 | 20 | 29 | 34.6 (0.93) | 32.29, 36.95 | 29.5 | 35 (20–50) |
| Beginning or ceasing formal schooling | 27 | 26 | 30 | 33.88 (0.91) | 31.57, 36.14 | 31.07 | 40 (25–55) |
| Major change in sleeping habits | 38 | 16 | 31 | 31.92 (0.84) | 29.83, 34.3 | 24.85 | 30 (20–40) |
| Outstanding personal achievement | 25 | 28 | 32 | 30.94 (0.96) | 28.49, 33.55 | 30.56 | 38.5 (21.3–50) |
| In-law troubles | 24 | 29 | 33 | 30.94 (0.91) | 28.62, 33.26 | 31.31 | 40 (25–60) |
| Troubles with the boss | 30 | 23 | 34 | 29.15 (0.9) | 26.72, 31.74 | 15.66 | 20 (10–30) |
| Major change in usual type and/or amount of recreation | 34 | 19 | 35 | 29.08 (0.78) | 27.01, 31.04 | 11.43 | 15 (5–30) |
| Major change in eating habits | 40 | 15 | 36 | 27.39 (0.77) | 25.2, 29.47 | 15.72 | 20 (10–30) |
| Taking on a loan for a lesser purchase | 37 | 17 | 37 | 25.12 (0.84) | 22.97, 27.6 | 17.36 | 20 (10–35) |
| Major change in social activities | 36 | 18 | 38 | 24.39 (0.8) | 22.31, 26.45 | 17.35 | 20 (10–30) |
| Major change in number of family get-togethers | 39 | 15 | 39 | 22.88 (0.76) | 20.9, 24.92 | 20.74 | 25 (10–40) |
| Revision of personal habits | 29 | 24 | 40 | 22.8 (0.77) | 20.95, 24.85 | 30.96 | 40 (25–53.8) |
| Minor violations of the law | 43 | 11 | 41 | 22.14 (0.76) | 20.12, 24.19 | 14.99 | 20 (10–30) |
| Major change in religious activities | 35 | 19 | 42 | 20.09 (0.8) | 18.01, 22.2 | 22.07 | 30 (15–40) |
| Vacation | 41 | 13 | 43 | 20.09 (0.81) | 18.03, 22.25 | 12.64 | 10 (6–30) |

*(Continued)*

**Table 8.** (Continued)

| Event | Holmes & Rahe | | Present Study | | | | |
|---|---|---|---|---|---|---|---|
| | rank | weight | rank | Mean (SE)[a] | 99% CI[b] | Geometric Mean[c] | Mdn (IQR)[d] |
| Christmas | 42 | 12 | 44 | 19.78 (0.84) | 17.79, 21.93 | 11.8 | 10 (5–30) |

Table ordered by Present Study's ranks, including 'Single person, living alone'.

[a] Mean (SE). Standard error obtained via BCa Bootstrap with 1000 samples.

[b] 99% confidence intervals obtained via BCa Bootstrap with 1000 samples.

[c] Where participants responded with a zero value, these were replaced with '1' to allow this value to be calculated.

[d] Mdn (IQR) = Median and inter-quartile range in brackets.

evidence supporting the null regarding YA vs. MA and MA vs. OA (BFs $\leq$ 0.17). Evidence comparing YA and OA was anecdotal (BF = 0.43). Comparing males and females, a Mann-Whitney U test revealed a statistically significant difference (z = -2.086, p = .034) with females (Mdn 40) rating this item higher than males (Mdn 30). The Bayesian Mann-Whitney U revealed anecdotal evidence (BF = 0.52), however. For ethnicity (white vs. non-white) and employment (employed vs. unemployed) ratings between groups were comparable for both frequentist (p > .3) and Bayesian (BF $\leq$ 0.14) tests. In contrast, both relationship status (married vs. unmarried) and religion (religion vs. no-religion) frequentist Mann Whitney tests revealed statistically significant outcomes. The married group (Mdn 40) assigned a higher rating than the unmarried group (Mdn 35) (z = -2.578, p = .01) and for the religion comparison, the religion group (Mdn 40) (z = -2.615, p = .009) rated this item higher than the no-religion group (Mdn 30). However, the corresponding Bayesian Mann-Whitney U results revealed anecdotal evidence for both relationship status (BF = 1.77) and religion (BF = 1.68). Thus, it remains unclear if participants from these respective sub-groups differ systematically regarding these items.

These results indicate that ratings for 'Single person, living alone' were comparable across age groups, ethnicity and employment status. However, for sex, religion and relationship status, evidence was inconclusive. Bayes factor sensitivity analyses (S6 Appendix) were comparable for all Mann-Whitney U comparisons reported above.

**Items proposed by respondents.** Participants were asked to provide a new item along with a corresponding rating for the amount of adjustment it required. Of 540 participants 259 (48%) gave no response, 50 (9.3%) suggested an item that was already in the SRRS. Further, one respondent gave a comment rather than an item, leaving 230 (42.6%) responses. Of 230 responses 43 items were 'one-off' suggestions (e.g. 'Brexit'), which were grouped as 'other' and a further 11 participants offered more than one item but with only one rating. These 54 responses were excluded from consideration because their respective ratings were either unclear or could not be averaged. The final list comprised 176 respondents' proposed new items along with their weights, given in Table 10. Items were given in participants' own words therefore item wording was chosen as appropriate. For example, the item: 'Death of a pet' was based on statements including 'loss of a pet', 'losing a pet', 'death of a family pet' and 'death of a pet'. The top 3 items were 'Mental health issue' (17%), 'Death of a pet' (14.8%) and 'Emigration' (8.5%). The averaged weights were 77, 72 and 69, respectively, as the table shows.

## Discussion

The SRRS weights were successfully updated using the ratings of 540 predominantly UK respondents aged 18 to 84. In addition, item wording was modernised, one optional extra item was added to the end of the scale and 3 potential new items proposed by raters were identified,

**Table 9. Descriptive statistics for 'Single person, living alone'.**

| | sub-groups | Mean (SE) | Median (IQR) |
|---|---|---|---|
| Age | | | |
| | < 30 years | 34.96 (2.61) | 30 (10–50) |
| | 30 to 60 years | 37.84 (1.52) | 35 (18–50) |
| | > 60 years | 41.66 (2.31) | 45 (20–65) |
| Sex | | | |
| | female | 40.1 (1.51) | 40 (20–60) |
| | male | 35.37 (1.73) | 30 (10–50) |
| Ethnicity | | | |
| | white | 38.62 (1.25) | 40 (15–52) |
| | non-white[a] | 35.72 (2.81) | 30 (15.5–50) |
| Religion | | | |
| | no religion | 35.35 (1.67) | 30 (10–50) |
| | religious[b] | 40.88 (1.56) | 40 (20–60) |
| Relationship status | | | |
| | married[c] | 40.22 (1.41) | 40 (20–60) |
| | unmarried[d] | 34.57 (1.93) | 35 (10–50) |
| Employment status | | | |
| | Employed | 37.26 (1.42) | 35 (15–50) |
| | unemployed[e] | 39.79 (1.93) | 40 (15.3–60) |
| | | | |

[a] 'non-white' included all ethnicities: mixed race, Asian (southern/southeastern Asia), Chinese (east Asian), black (any region).

[b] 'religious' includes all religions: Christian, Buddhist, Hindu, Jewish, Muslim, Sikh, any other religion.

[c] 'married' includes married and long-term partners.

[d] 'unmarried' includes those who don't qualify as 'c': divorced, in a relationship, life partner died, separated, single, widowed.

[e] 'unemployed' includes those currently unemployed and looking for work, have never worked, long-term sick/disabled, looking after home or family, retired, student (p/t or f/t) and currently unemployed, other.

namely: 'Mental health issue', 'Death of a pet' and 'Emigration'. Changes/modernisations made did not affect the meaning of any of the original items. No items were removed. By doing so, the SRRS was improved and backwards-compatibility with the original scale, that continues to be widely used, was maintained.

The main findings were that the updated SRRS yielded a significantly higher total score on average than the original scale. However, the new weights were broadly consistent with the original weights, indicating continuity between US and UK samples. Comparing all life events in the original and new scales using Rahe's [24] 4 categories, 'personal', 'family', 'work' and 'finance' it was found, as with the overall correlation, that there was a strong covariance for all categories except finance. Focusing on the present study's sample, young, middle-aged and older adults were comparable in their total LCU scores. Females assigned, on average, 14% higher weight to life events than males but the statistical evidence for this gender-based variation in life change units was inconclusive. Similarly, while the sample's +70s group assigned, on average, a 4% lower weight to life events than the normative-aged sample, this difference was statistically negligible. Regarding the possible influence of personal experience on ratings, some significant associations were found, though coefficient sizes were very small. The new additional item, 'Single person, living alone' required less adjustment relative to marriage. Most commonly, participants felt lonely occasionally and average level of loneliness was rated towards the lower end of the range. Loneliness was, at best, weakly associated with the SRRS

**Table 10. Mean and standard errors (SE) for own items and frequency of raters per item.**

| Suggested additional items for SRRS | Number of raters (n = 176) | Mean (SE) |
|---|---|---|
| Mental health difficulties | 30 | 77.2 (3.56) |
| Death of pet | 26 | 72.31 (4.43) |
| Emigration | 15 | 69.27 (4.93) |
| Relationship break-up | 12 | 71.25 (6.8) |
| Covid (having the illness/unspecified) | 10 | 76.5 (7.15) |
| Covid restrictions (e.g. lock-down) | 7 | 70 (7.94) |
| Accident (e.g. car accident) | 6 | 57.17 (10.59) |
| Becoming a carer (e.g. elderly relative) | 6 | 81.67 (5.43) |
| Getting a pet | 6 | 38.33 (4.94) |
| Infidelity (having affair) | 6 | 78.67 (7.23) |
| Relocation | 5 | 62 (3.39) |
| War/conflict | 5 | 95 (5) |
| Addiction | 4 | 62.5 (8.54) |
| Change in state policy/regime (e.g. Brexit) | 4 | 60.75 (14.08) |
| Natural disaster | 4 | 80 (4.08) |
| Sexuality/gender-identity (e.g. identifying as gay) | 4 | 57.5 (12.5) |
| Trouble with neighbours | 4 | 48.75 (14.35) |
| Victim of crime | 4 | 48.75 (14.78) |
| Abuse | 3 | 88.33 (4.41) |
| Adjusting to older age | 3 | 60 (5.77) |
| Bullying | 3 | 75 (2.89) |
| Domestic violence | 3 | 91.67 (4.41) |
| Assault | 2 | 47.5 (17.5) |
| Terminal illness | 2 | 82.5 (7.5) |
| Wedding | 2 | 80 (20) |

ratings given. Thus, participants' ratings, particularly for 'Single person, living alone' were unlikely to have been influenced by their experience of loneliness.

## SRRS life events require more adjustment now than in 1967

A comparison with the original SRRS weights indicated an overall average increase of 28%, which suggests that, on average, adults find SRRS events more taxing now than in 1967. Consistent with this increase, Miller and Rahe [13]'s update (n = 426) revealed a 45% overall increase. However, as in their update, a strong covariance pattern across ratings was found, but with some exceptions. Three of the items increased by $\geq 25$ life change units (LCUs) relative to the original scale. They were also among the 6 items that had increased by $\geq 25$ LCUs in the Miller and Rahe [13] update. Taken together, the aforementioned outcomes suggest that since the original weights were derived there has been considerable change in perceived adjustment needed regarding certain key life events such as pregnancy and foreclosure on a loan, but on the whole, there is overall consensus regarding relative importance of the life events in the scale. The exception being financial items. The new sample's weights for the 4 financial life events did not correlate with the original sample and, as a category, showed the largest average increase in weights while family items showed the smallest increase. By comparison, the Miller and Rahe [13] update also showed the largest average increase in weights for financial items but the smallest increase was for personal items. Interestingly, the Scully et al. [10] update, relative to the original, indicated that across categories, weights consistently decreased but the

smallest decrease was for financial items. Scully's sample comprised 200 Florida residents, however, which makes their result less generalisable. The relative volatility of the financial items may be because there are only 4 items included in this sub-set. In addition, one could argue that economic factors are generally more volatile than social factors though a thorough examination of this would be required.

### Females' ratings for SRRS life events were slightly higher

Females' total LCUs were 14% higher on average relative to males' LCUs, though this difference was inconclusive. However, an in-depth Bayesian analysis for each of the 4 categories revealed substantial to very strong evidence that females' weights were consistently higher than males' weights. This pattern of higher weights among females is in line with Miller and Rahe [13]'s update reporting that their females' weights were 17% higher. They did not report whether this difference was statistically significant. This pattern indicating that females typically assign a greater level of adjustment to change relative to males, warrants more detailed investigation in future work. Research investigating the gender-specific profiles of psychiatric disorders indicate that stressed women become hyper-aroused, which is a common feature of depression [44]. However, stressed men's cognitive function can be differentially disrupted [44,45] and evidence shows that these differences associated with stress may be consequent to sex-based differences within the locus coeruleus-norepinephrine arousal system [44,46]. These results show that there are numerous underlying psychological, physiological and environmental factors to consider regarding sex-based differences in stress-reactivity and, by implication, perceived adjustment to different life events.

### Adults aged 70+ rate SRRS items similarly to adults aged 18 to 69

The normative and 70+ groups gave comparable ratings on average. In contrast, Muhlenkamp et al.'s [12] study found that older participants assigned higher weights relative to the original 1967 sample. However, in line with the present study's results, they concluded that there was agreement regarding relative importance of some life events. They argued that their older group's higher ratings for 'personal' life events were consistent with literature indicating that adults become more self-orientated (egocentric) with age [12]. By comparison, the present results are congruent with the consistent finding in the literature that social and emotional functioning does not change significantly across the lifespan and that self-regulation, including coping with stressors, improves with age [47]. Further support of this is that the young, middle-aged and older adults were comparable in their ratings, which yielded very similar averages and age did not reveal any systematic differences when considering items as categories. However, this does not mean that age should be ignored. While older age is associated with improved emotional well-being, they respond less well compared to the young in some situations. For example, older adults show more pronounced psychological and physiological reactions relative to younger adults when stressful events are complex, affecting multiple life domains [48]. It is also worth noting that middle adulthood is associated with increased roles and responsibilities such as career progression and having a family. Moreover, they are pivotal to the younger and older members of their family [49] which can be very stressful particularly for those caring for children and elderly parents/relatives [50]. These points highlight the complexity of measuring the impact of life events at different points across the lifespan.

### Adding ratings for a new item: 'Single person, living alone'

'Single person, living alone', the new item, yielded some differences in ratings between age groups with the greatest disparity between young and older adults (OA>YA). However, the

result was inconclusive. The same was true for comparisons by relationship status, religion and sex. For ethnicity and employment status evidence supported the null. Particularly regarding the age-related finding, this life event is important to monitor in future work. The ONS [51] data indicate a continuing upward trend in the number of one-person households. Moreover, the ONS data agrees with the findings of a recent study showing that among those in western countries, such as Europe and North America, more males than females live alone among young (25 to 29) and middle-aged adults (50 to 54) while more females live alone among older adults (75 to 79) [52]. Their review of global patterns further revealed that family-based living is most common, except in older age. Indeed, among ageing populations around the world, the number of older adults living alone is likely to increase (ibid). It is important to distinguish between sub-populations of older adults who live alone, as those with a diverse social network (e.g. adult children, siblings, friends and neighbours to socialise with and ask for support) have been shown to have better well-being than those with a restricted network (few family and friends/neighbours to reach out to) [53].

### Raters' 3 new items: 'Mental health issue', 'Death of a pet', 'Emigration'

Roughly half the respondents added an additional item to the given list of events to rate. The 3 most common items are indicated in the sub-title above. The full table of items provides further opportunities for future investigation.

### The impact on ratings of personal experience and loneliness

No clear pattern of association was found between ratings and personal experience of the life events nor between ratings and loneliness scores. Regarding personal experiences, 35% of items were significantly correlated but coefficient sizes were of no practical significance, suggesting that respondents were not unduly influenced by their own personal experiences (i.e. they adhered to the instructions to consider their own experiences as well as those of others). The results for loneliness revealed that for the 2 items, 'Revision of personal habits' and 'Single person, living alone', there was a significant association with loneliness. Again, however, the coefficients were very small, providing confidence that ratings were not biased by loneliness scores suggesting that ratings did not vary systematically with emotional state.

### Limitations

A convenience sample rather than a randomly selected sample was used, which may have introduced selection bias. One could speculate that those who completed the survey were relatively more pro-active because participation required time and effort and had more resources because they would need basic computer skills, access to the internet and an internet-accessible device and time. Based on these assumptions, bias would be introduced by not including the segment of the UK population who were too stressed or struggling either financially or in other ways to take part. Their ratings may have been somewhat higher compared to our sample's for items such as 'Foreclosure/repossession on mortgage or loan', 'Detention in jail or other institution' and 'Losing your job'. Using the present study's updated weights are likely suitable for the general population but if researchers intend to use these with specific sub-populations such as trauma survivors or those living in severe poverty, additional validation is advised. The same is true regarding ethnic and religious minority groups who represent a relatively small proportion within the UK population. Secondly, the updated weights agree broadly with the original study based on a US sample and would therefore probably generalise to populations culturally similar to the UK and US. Care should therefore be taken when using the updated weights with non-Western cultures. In addition, future work could include more extended differential item

functioning analysis in these subgroups to explore the universality of stressful life events and their ratings. Thirdly, the statistical power was sub-optimal for between-groups comparisons, based on the Gpower calculations. However, Bayesian analyses together with accompanying sensitivity analyses provides valid statistical evidence for the findings reported.

## Conclusion

This study's aim was to update and improve the SRRS weights without fundamentally changing the scale to allow for backwards-compatibility of studies. This was achieved by providing new weights for all original items. The updated weights were higher but broadly consistent with those of the original study except for financial items, which may vary considerably over time. Cross-comparability with the original version was allowed for by retaining all original items, making helpful changes to the wording of some items without changing their meaning and adding a new item to the end, which can be excluded for comparisons. This sample's ratings were not systematically influenced by personal experiences of events or loneliness. Such factors are important indicators of consistency across ratings but were not considered in the original scale nor subsequent versions.

The present study provides updated weights derived from a predominantly UK sample which is broadly proportionately representative regarding age, gender and ethnicity. Moreover, the age-range was broader and sample size slightly larger than the original. Though it is noted that this study was conducted using a convenience sample with a modest sample size. A new item was added to the end of the scale, 'Single person, living alone', which will benefit future work given that single-person households have increased in recent years. Three potential new items submitted by respondents have been identified: 'Mental health issue', 'Death of a pet' and 'Emigration'. Further work may be undertaken in future studies to determine whether these items would be beneficial to add. What is known about the impact of demographics, such as gender and age, on the rating of life events was updated and extended, providing additional scope for future work.

## Supporting information

**S1 Checklist. The SRRS updated and modernised: STROBE checklist.**
(DOCX)

**S1 Appendix. SRRQ (updated version).**
(PDF)

**S2 Appendix. Original Instructions for SRRS.**
(PDF)

**S3 Appendix. SRRS updated version.**
(PDF)

**S4 Appendix. Personal experience of SRRS life events.**
(PDF)

**S5 Appendix. Loneliness questionnaire.**
(PDF)

**S6 Appendix. Bayes Factor sensitivity analysis for all Mann-Whitney U comparisons.**
(PDF)

**S7 Appendix. Social Readjustment Rating Scale by category.**
(PDF)

**S8 Appendix. Descriptive statistics for SRRS events by unabridged sub-group demographics.**
(PDF)

**S9 Appendix. Bayesian Kendall's tau correlation between event ratings and degree to which these were based on personal experience.**
(PDF)

## Author Contributions

**Conceptualization:** Denise Wallace.

**Data curation:** Denise Wallace.

**Formal analysis:** Denise Wallace.

**Funding acquisition:** Denise Wallace.

**Investigation:** Denise Wallace.

**Methodology:** Denise Wallace, Nicholas R. Cooper, Alejandra Sel, Riccardo Russo.

**Project administration:** Denise Wallace.

**Resources:** Denise Wallace.

**Software:** Denise Wallace.

**Supervision:** Nicholas R. Cooper, Alejandra Sel, Riccardo Russo.

**Validation:** Denise Wallace, Nicholas R. Cooper, Alejandra Sel, Riccardo Russo.

**Visualization:** Denise Wallace.

**Writing – original draft:** Denise Wallace.

**Writing – review & editing:** Denise Wallace, Nicholas R. Cooper, Alejandra Sel, Riccardo Russo.

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
