## [Decision Letter · Decision Letter 0]

18 Oct 2023

PONE-D-23-17307The Social Readjustment Rating Scale: updated and modernisedPLOS ONE

Dear Dr. Wallace,

Thank you for submitting your manuscript to PLOS ONE. After careful consideration, we feel that it has merit but does not fully meet PLOS ONE’s publication criteria as it currently stands. Therefore, we invite you to submit a revised version of the manuscript that addresses the points raised during the review process.

We look forward to receiving your revised manuscript.

Kind regards,

Adetayo Olorunlana, Ph.D.

Academic Editor

PLOS ONE

Journal Requirements:

Reviewers' comments:

Reviewer's Responses to Questions

**Comments to the Author**

1. Is the manuscript technically sound, and do the data support the conclusions?

Reviewer #1: Yes

Reviewer #2: Yes

2. Has the statistical analysis been performed appropriately and rigorously? 

Reviewer #1: Yes

Reviewer #2: Yes

3. Have the authors made all data underlying the findings in their manuscript fully available?

Reviewer #1: Yes

Reviewer #2: Yes

4. Is the manuscript presented in an intelligible fashion and written in standard English?

Reviewer #1: Yes

Reviewer #2: Yes

5. Review Comments to the Author

Reviewer #1: Dear author, thank you for the opportunity to review your manuscript. I can say that the main goals of this study are accomplished. There are good results, highlighted from the findings. The methodology is well described and the research questions are clear. Moreover, statistical analyses is well conducted and the literature is carefully selected. I can say that the manuscript is well revised and it is ready for the publication. Thank you.

Reviewer #2: Dear authors,

Thank you for giving me the opportunity to review your manuscript. Frankly, it was a joy to read. I have worked with the old version of the Holmes & Rahe scale and had wondered when somebody would manage to update this.

I only have minor comments to make.

(a) I understand that you have essentially done a quota sampling approach. While this can be representative as a sampling approach to an extent, it is not a random sampling approach per se. Particularly regarding the content validity of stressful life events and whether ratings can be extrapolated to the whole population, I was wondering whether it would have been worthwhile to

- oversample certain ethnic and minority groups

- do a more extended DIF analysis in these subgroups to understand further whether both stressful life events and their ratings are truly universal like you seem to imply.

I understand that this comment refers to a different sampling approach than the one you have done. I'd appreciate if you could go into more detail on problems regarding the universality of the ratings, particularly on certain minority ethnic groups. Given that the UK population is multi-cultural (as other societies are), it would pay to think through how to ensure better validity and representation for all groups involved. It would also create further discussions and might give you ideas for further research studies.

(b) I would appreciate some more thoughts on using marriage as the anchor. Why have you done so, given that marriage in our society does not hold the same meaning than it did when the original version of the scale was developed? What does it mean for your ratings that you picked an anchor item with quite a low rating?

As a knock-on effect - what does it mean that the most stressful event is not anchored to a 100% anymore?

I have a personal preference for using item response theory to better understand item difficulty. Might this be an approach you could take to reanalyse the data to better understand the respective item difficulty and the corresponding person abilities in your sample?

6. PLOS authors have the option to publish the peer review history of their article (what does this mean?). If published, this will include your full peer review and any attached files.

Reviewer #1: No

Reviewer #2: No

---

## [Author Response · Author response to Decision Letter 0]

29 Nov 2023

MS # PONE-D-23-17307: Our response to reviewer comments

Dear Editor, 

Thank you for the opportunity to re-submit our revised manuscript.

Our submission includes a rebuttal letter and a marked up and unmarked copy of our manuscript. We address specific points below.

We would like to thank the reviewers for their time and thoughtfulness in reviewing our manuscript. The feedback from both reviewers is gratefully received. 

To Reviewer 1, thank you for giving our study a clean bill of health. We really appreciate receiving such positive feedback from a reviewer.

To Reviewer 2, thank you for your positive feedback and insightful suggestions for future follow-up studies. We reply to each comment point by point below:

Reviewer 2 Point 1: 

(a) I understand that you have essentially done a quota sampling approach. While this can be representative as a sampling approach to an extent, it is not a random sampling approach per se. Particularly regarding the content validity of stressful life events and whether ratings can be extrapolated to the whole population, I was wondering whether it would have been worthwhile to

- oversample certain ethnic and minority groups

- do a more extended DIF analysis in these subgroups to understand further whether both stressful life events and their ratings are truly universal like you seem to imply.

I understand that this comment refers to a different sampling approach than the one you have done. I'd appreciate if you could go into more detail on problems regarding the universality of the ratings, particularly on certain minority ethnic groups. Given that the UK population is multi-cultural (as other societies are), it would pay to think through how to ensure better validity and representation for all groups involved. It would also create further discussions and might give you ideas for further research studies.

Reply to Point 1: 

Thank you for raising this very important point regarding representation. The purpose of our study was to update the SRRS which is a population-based measure. Our study was therefore designed to represent (proportionately) the UK’s current population regarding age, gender and ethnicity. Researchers can and do use the SRRS in other contexts and should note the point regarding representation which we discuss in our limitations section. We agree that research questions concerning differences between ethnic, cultural and religious groups would add value to the understanding of life events stress. I have made two amendments in the Limitations section noting that some groups within the UK population represent a relatively small proportion of the UK population and to recommend more in-depth DIF analysis in future work:

[lines 870-871] “…living in severe poverty, additional validation is advised. The same is true regarding ethnic and religious minority groups who represent a small proportion within the UK population.”

[lines 875-877] “…updated weights with non-Western cultures. In addition, future work could include more extended differential item functioning analysis in these subgroups to explore the universality of stressful life events and their ratings.”

Reviewer 2 Point 2: 

(b) I would appreciate some more thoughts on using marriage as the anchor. Why have you done so, given that marriage in our society does not hold the same meaning than it did when the original version of the scale was developed? What does it mean for your ratings that you picked an anchor item with quite a low rating?

As a knock-on effect - what does it mean that the most stressful event is not anchored to a 100% anymore?

Reply to Point 2: 

It is fundamentally important to provide an update that is compatible with the original version of the SRRS otherwise researchers may not want to use it. For this reason, we used a proportionate scaling approach with marriage as an anchor to align our update with the original Holmes and Rahe (1967) SRRS. Marriage was not a rated item in the original scale, the value was pre-assigned and so, our approach is in line with the original one taken by Holmes and Rahe.

Regarding the knock-on effect, we used marriage as the anchor point. ‘Death of a spouse/life-partner’ while no longer at 100 life change units is still ranked 1st. Its relative position remains unchanged.

Reviewer 2 Point 3: 

I have a personal preference for using item response theory to better understand item difficulty. Might this be an approach you could take to reanalyse the data to better understand the respective item difficulty and the corresponding person abilities in your sample?

Reply to Point 3: 

The design and purpose of this study was to update and improve the SRRS in a backwards-compatible way. We agree that the issue of item difficulty is a valid concern with survey research. This research question would be important to address in a future study.

Once again, our thanks to both reviewers for their valued contribution to our manuscript.

With kind regards,

Denise Wallace

On behalf of the authors

---

## [Decision Letter · Decision Letter 1]

4 Dec 2023

The Social Readjustment Rating Scale: updated and modernised

PONE-D-23-17307R1

Dear Dr. Wallace,

We’re pleased to inform you that your manuscript has been judged scientifically suitable for publication and will be formally accepted for publication once it meets all outstanding technical requirements.

Kind regards,

Adetayo Olorunlana, Ph.D.

Academic Editor

PLOS ONE

Additional Editor Comments (optional):

Reviewers' comments:

Reviewer's Responses to Questions

**Comments to the Author**

1. If the authors have adequately addressed your comments raised in a previous round of review and you feel that this manuscript is now acceptable for publication, you may indicate that here to bypass the “Comments to the Author” section, enter your conflict of interest statement in the “Confidential to Editor” section, and submit your "Accept" recommendation.

Reviewer #2: All comments have been addressed

2. Is the manuscript technically sound, and do the data support the conclusions?

Reviewer #2: Yes

3. Has the statistical analysis been performed appropriately and rigorously? 

Reviewer #2: Yes

4. Have the authors made all data underlying the findings in their manuscript fully available?

Reviewer #2: Yes

5. Is the manuscript presented in an intelligible fashion and written in standard English?

Reviewer #2: Yes

6. Review Comments to the Author

Reviewer #2: Thank you for providing such a thoughtful reply. All my comments have been addressed. This was a joy to re-review.

7. PLOS authors have the option to publish the peer review history of their article (what does this mean?). If published, this will include your full peer review and any attached files.

Reviewer #2: **Yes: **Christina Ramsenthaler

---

## [Editor Report · Acceptance letter]

7 Dec 2023

PONE-D-23-17307R1 

The Social Readjustment Rating Scale: updated and modernised 

Dear Dr. Wallace:

I'm pleased to inform you that your manuscript has been deemed suitable for publication in PLOS ONE. Congratulations! Your manuscript is now with our production department. 

Kind regards, 

on behalf of

Associate Professor Adetayo Olorunlana 

Academic Editor

PLOS ONE